# Beyond Expected Returns: A Policy Gradient Algorithm for Cumulative Prospect Theoretic Reinforcement Learning

## Abstract

The widely used expected utility theory has been shown to be empirically inconsistent with human preferences in the psychology and behavioral economy literatures. Cumulative Prospect Theory (CPT) has been developed to fill in this gap and provide a better model for human-based decision-making supported by empirical evidence. It allows to express a wide range of attitudes and perceptions towards risk, gains and losses. A few years ago, CPT has been combined with Reinforcement Learning (RL) to formulate a CPT policy optimization problem where the goal of the agent is to search for a policy generating long-term returns which are aligned with their preferences. In this work, we revisit this policy optimization problem and provide new insights on optimal policies and their nature depending on the utility function under consideration. We further derive a novel policy gradient theorem for the CPT policy optimization objective generalizing the seminal corresponding result in standard RL. This result enables us to design a model-free policy gradient algorithm to solve the CPT-RL problem. We illustrate the performance of our algorithm in simple examples motivated by traffic control and electricity management applications. We also demonstrate that our policy gradient algorithm scales better to larger state spaces compared to the existing zeroth order algorithm for solving the same problem.

## 1 Introduction

In classical reinforcement learning (RL), rational agents make decisions to maximize their expected cumulative rewards through interaction with their environment. This paradigm has largely been prescribed by the expected utility theory model which has dominated decision making. Besides this risk-neutral setting, risk-seeking and risk-averse behaviors can also be individually modelled within the same expected utility maximization paradigm by considering the expectation of a modified utility function as a policy optimization objective (see e.g. Prashanth et al. (2022) for a recent survey).

However, human decision makers might not act rationally due to psychological biases and personal preferences, their decisions might not necessarily be dictated by expected utility theory. Consider this simple example as a first illustration: A player must choose between (A) receiving a payoff of 80 and (B) participating in a lottery and receive either 0 or 200 with equal probability. The player's preference depends on their attitude towards risk. While a risk-neutral agent will be satisfied with the immediate and safe payoff of 80, another individual might want to try to obtain the much higher 200 payoff. In particular, different agents might perceive the same utility and the same random outcome differently. Furthermore, they can exhibit both risk-seeking and risk-averse behaviors depending on the context. Therefore, due to its failure to capture such settings as a descriptive model, the standard expected utility theory has been called into question by the pioneering behavioral psychologist Daniel Kahneman together with his colleague Amos Tversky (Kahneman & Tversky, 1979). In particular, Daniel Kahneman has been awarded the Nobel Prize in Economic Sciences in 2002 "for having integrated insights from psychological research into economic science, especially concerning human judgment and decision-making under uncertainty". In their seminal works combining cognitive psychology and economics, they laid the foundations of the so-called prospect theory and its cumulative version later on (Tversky & Kahneman, 1992) to explain several empirical observations invalidating the standard expected utility theory. Let us illustrate this in a simple example borrowed from Ramasubramanian et al. (2021) (example 2 in section IV therein) for the purpose of

our exposition. Consider a game where one can either earn \$100 with probability (w.p.) 1 or earn 10000 w.p. 0.01 and nothing otherwise. A human might rather lean towards the first option which gives a certain gain. In contrast, if the situation is flipped, i.e., a loss of 100 w.p. 1 versus a loss of \$10000 w.p. 0.01, then humans might rather choose the latter option. In both settings, the expected gain or loss has the same value (100). The CPT paradigm allows to model the tendency of humans to perceive gains and losses differently. Moreover, the humans tend to deflate high probabilities and inflate low probabilities (Tversky & Kahneman, 1992; Barberis, 2013). For instance, as exposed in L.A. et al. (2016), humans might rather choose a large reward, say 1 million dollars w.p. $10^{-6}$ over a reward of 1 w.p. 1 and the opposite when rewards are replaced by losses.

Inspired by Kahneman and Tversky's findings, CPT has been used in a number of applications in the stateless setting such as energy retrofit decision for home renovations (Ebrahimigharehbaghi et al., 2022) and smart home energy management Dorahaki et al. (2022), building evacuation (Gao et al., 2023), shared parking services (Yan et al., 2020) and financial decision making (Ladrón de Guevara Cortés et al., 2023; Luxenberg et al., 2024) to name a few. We refer the reader to appendix C for an extended discussion regarding applications. Recently, a line of research initiated by L.A. et al. (2016) has combined CPT with RL to better account for the human behavior in decision making (Borkar & Chandak, 2021; Ramasubramanian et al., 2021; Danis et al., 2023; Ethayarajh et al., 2024). As highlighted in Borkar & Chandak (2021), this is particularly important in applications directly involving humans in the loop such as e-commerce, crowdsourcing and recommendation to name a few. As empirically demonstrated and discussed in Tversky & Kahneman (1992), CPT allows to capture two specific features of human decision making: Humans tend to (a) be risk-seeking with potential losses and risk-averse with possible gains, this is modelled via using an S-shaped non-linear transformation of the utility function; (b) overestimate the probability of rare events and underestimate the probability of frequent events. CPT uses for this a weighting function to distort the cumulative probability distribution function, inflating low probabilities and deflating high probabilities. In this work, we focus on the policy optimization problem where the objective is the CPT value of the cumulative sum of rewards, induced by a parametrized policy in a Markov Decision Process. Our main contributions are as follows:

**About optimal policies in CPT-RL.** We provide theoretical insights about the nature of an optimal policy for CPT policy optimization. Unlike in standard MDPs, an optimal policy is stochastic and non-Markovian in general. When we set the probability distortion function to identity, we show that the policy search set can be significantly reduced to a much smaller policy class when solving (CPT-PO). In this same setting, we also characterize a family of utility functions (affine and exponential utility functions) for which the CPT value objective can be maximized with a Markovian policy. However, we prove that this characterization does not hold anymore when considering nontrivial probability distortion and (nonlinear) utilities together in (CPT-PO).

**Policy gradient theorem and algorithm for CPT-RL.** We establish a policy gradient theorem providing a closed form expectation expression for the gradient of our CPT-value objective w.r.t. the policy parameter under suitable regularity conditions on the utility and probability distortion functions. This result generalizes the standard policy gradient theorem in RL. Building on this theorem, we design a policy gradient algorithm to solve the CPT policy optimization problem. The stochastic policy gradient we use involves a challenging integral term to be computed and we propose a tailored estimation procedure to approximate it.

**Experiments.** We perform simulations to illustrate our theoretical results on simple examples. In particular, we test our PG algorithm in two CPT-RL applications: a traffic control application with finite discrete state action spaces and an electricity management task in a continuous state action setting. We also compare the performance of our PG algorithm to the previously proposed zeroth order algorithm to show the robustness and scalability of our algorithm to higher dimensional MDPs.

## 2 PRELIMINARIES: FROM CLASSICAL RL TO CPT-RL

**Markov Decision Process.** A discrete-time discounted Markov Decision Process (MDP) (Puterman, 2014) is a tuple $\mathcal{M} = (\mathcal{S}, \mathcal{A}, \mathcal{P}, r, \rho, \gamma)$, where $\mathcal{S}, \mathcal{A}$ are respectively the state and action spaces, supposed to be finite for simplicity, $\mathcal{P} : \mathcal{S} \times \mathcal{A} \times \mathcal{S} \to [0, 1]$ is the state transition probability kernel, $r : \mathcal{S} \times \mathcal{A} \to [-r_{\max}, r_{\max}]$ is the reward function which is bounded by $r_{\max} > 0$, $\rho$ is the initial state probability distribution, and $\gamma \in (0, 1)$ is the discount factor. A randomized stationary

Markovian policy, which we will simply call a policy, is a mapping $\pi : \mathcal{S} \to \Delta(\mathcal{A})$ which specifies for each $s \in \mathcal{S}$ a probability measure over the set of actions $\mathcal{A}$ by $\pi(\cdot|s) \in \Delta(\mathcal{A})$ where $\Delta(\mathcal{A})$ is the simplex over the finite action space $\mathcal{A}$. Each policy $\pi$ induces a discrete-time Markov reward process $\{(s_t, r_t := r(s_t, a_t))\}_{t \in \mathbb{N}}$ where $s_t \in \mathcal{S}$ represents the state of the system at time $t$ and $r_t$ corresponds to the reward received when executing action $a_t \in \mathcal{A}$ in state $s_t \in \mathcal{S}$. We denote by $\mathbb{P}_{\rho,\pi}$ the probability distribution of the Markov chain $(s_t, a_t)_{t \in \mathbb{N}}$ generated by the MDP controlled by the policy $\pi$ with initial state distribution $\rho$. We use $\mathbb{E}_{\rho,\pi}$ (or often simply $\mathbb{E}$ instead) to denote the expectation w.r.t. the distribution of the Markov chain $(s_t, a_t)_{t \in \mathbb{N}}$. At each time step $t \geq 0$, the agent follows its policy $\pi$ by selecting an action $a_t$ drawn from the action distribution $\pi_t(\cdot|s_t)$ where $s_t$ is the environment state at time $t$. Then the environment transitions to a state $s_{t+1}$ sampled from the state distribution $\mathcal{P}(\cdot|s_t, a_t)$ given by the state transition kernel $\mathcal{P}$ and the agent obtains a reward $r_t$. In traditional RL, the goal of the agent in discounted MDPs is to find a policy $\pi$ maximizing the expected cumulative discounted rewards, i.e. the so-called expected return $J(\pi) := \mathbb{E}_{\rho,\pi}[\sum_{t=0}^{H-1} \gamma^t r_t]$ where $s_0$ follows the initial state distribution $\rho$ and $H \geq 1$ is a finite horizon. Any fixed policy $\pi$ and any initial state distribution $\rho$ induce together a state occupancy measure $d_\rho^\pi$ recording the visitation frequency of each state, it is defined at each state $s \in \mathcal{S}$ by $d_\rho^\pi(s) := \sum_{t=0}^{H-1} \gamma^t \mathbb{P}_{\rho,\pi}(s_t = s)$. The corresponding state-action occupancy measure is defined for every state-action pair $(s, a) \in \mathcal{S} \times \mathcal{A}$ by $\mu_\rho^\pi(s, a) := d_\rho^\pi(s)\pi(a|s)$. Recall that $J(\pi) = \langle \mu_\rho^\pi, r \rangle := \sum_{s \in \mathcal{S}, a \in \mathcal{A}} \mu_\rho^\pi(s, a) r(s, a)$ for any policy $\pi$ and any initial state distribution $\rho$.

**Policy classes.** We now introduce different sets of policies which will be important for stating our results. Each policy class is defined according to the information history the policies have access to for selecting actions. Here, a history $h_t \in \mathcal{H}$ is a finite sequence of successive states, actions and rewards: $(s_0, a_0, r_0, ...., s_{t-1}, a_{t-1}, r_{t-1})$.[1] More specifically, throughout this work, we will consider the following sets of policies: $\Pi_{NM} := \{\mathcal{H} \to \Delta(\mathcal{A})\}$ is the set of non-(necessarily)Markovian policies, $\Pi_{\Sigma,NS} := \{\mathcal{S} \times \mathbb{R} \times \mathbb{N} \to \Delta(\mathcal{A})\}$ is the set of policies that only depend on the current state, the timestep and the sum of rewards accumulated so far (i.e. $\pi(s, \sum_{k=0}^{t-1} r_k, t)$), $\Pi_{\Sigma,S} := \{\mathcal{S} \times \mathbb{R} \to \Delta(\mathcal{A})\}$ is the set of policies that only depend on the state and the sum of rewards (i.e. $\pi(s, \sum_{k=0}^{t-1} r_k)$), $\Pi_{M,NS} := \{\mathcal{S} \times \mathbb{N} \to \Delta(\mathcal{A})\}$ is the set of Markovian policies (i.e. $\pi(s, t)$) and $\Pi_{M,S} := \{\mathcal{S} \to \Delta(\mathcal{A})\}$ is the set of stationary Markovian policies, i.e. Markovian policies which are time-independent. Deterministic policies assign a single action to each state. For each set of policies defined above, we define their corresponding subset of deterministic policies: $\Pi_{NM}^D, \Pi_{\Sigma,NS}^D, \Pi_{\Sigma,S}^D, \Pi_{M,NS}^D$ and $\Pi_{M,S}^D$. With some flexibility on the notation, deterministic policies can either be written as functions with values in $\Delta(\mathcal{A})$ like their nondeterministic counterparts, or directly as functions with values in $\mathcal{A}$.

**Remark 1.** $\Pi_{M,S} \subseteq \Pi_{M,NS} \subseteq \Pi_{\Sigma,NS} \subseteq \Pi_{NM}$ and $\Pi_{M,S} \subseteq \Pi_{\Sigma,S} \subseteq \Pi_{\Sigma,NS} \subseteq \Pi_{NM}$ (Fig. 4).

**Cumulative Prospect Theory Value.** Instead of the expected return, CPT prescribes to consider the CPT value which will be defined in this paragraph. As previously mentioned, CPT relies on three distinct elements which we further detail in the following:

**(a) A reference point.** The human agent has a reference attainable reward value in comparison to which they evaluate their possible reward outcomes. Rewards larger than the reference are perceived as gains whereas lower values are viewed as losses.

**(b) A utility function** $\mathcal{U} : \mathbb{R} \to \mathbb{R}_+$**.** The agent's utility is a continuous and non-decreasing function which is not necessarily linear w.r.t. the total reward received by the agent. We consider the function $u^+ : \mathbb{R} \to \mathbb{R}_+$ describing the gains and defined for every $x \in \mathbb{R}$ by $u^+(x) = \mathcal{U}(x)$ if $x \geq x_0$ and zero otherwise. Similarly, the function $u^- : \mathbb{R} \to \mathbb{R}_+$ which encodes the losses is defined by $u^-(x) = -\mathcal{U}(x)$ if $x \leq x_0$ and zero otherwise. Here, $x_0$ denotes the reference point. Typically, the utility function is concave (respectively convex) for positive (resp. negative) rewards w.r.t. the reference point, i.e. $u^+$ is concave on $\mathbb{R}_+$ and $-u^-$ is convex on $\mathbb{R}_-$. For concreteness, we will use Kahneman & Tversky (1979)'s utility function as a running example: $\mathcal{U}(x) = (x - x_0)^\alpha$ if $x \geq x_0$ and $\mathcal{U}(x) = -\lambda(x - x_0)^\alpha$ if $x < x_0$, where $\lambda = 2.25, \alpha = 0.88$ are recommended hyperparameters. See Fig. 6 for an illustration with $x_0 = 0$.

**(c) A probability distortion function** $w : [0, 1] \to [0, 1]$**.** This is a continuous non-decreasing weight function that distorts the probability distributions of the gain and loss variables. This func-

---

[1]Rewards can be discarded from the history when they are deterministic functions of state-action pairs.

tion typically captures the human tendency to overestimate the probability of rare events and underestimate the probability of more certain ones. Similarly to the utility function, we denote by $w^+$ (resp. $w^-$) the function that warps the cumulative distribution function for gains (resp.for losses). Both functions are required to be defined on $[0, 1]$, with values in $[0, 1]$ and to be non-decreasing, continuous, with $w^+(0) = w^-(0) = 0$ and $w^-(1) = w^-(1) = 1$. Examples of such weights functions in the litterature include $w : p \mapsto p^\eta (p^\eta + (1-p)^\eta)^{-\frac{1}{\eta}}$ (Kahneman & Tversky (1979)) and $w : p \mapsto \exp(-(-\ln(p)^\eta))$ (Prelec (1998)) where $\eta \in (0, 1)$ is a hyperparameter. We refer the reader to appendix E.2 for examples and plots of utility and probability weight functions.

Following the exposition in L.A. et al. (2016), we use the notation $\mathbb{C}(X)$ to denote the CPT value of a real-valued random variable $X$:

$$\mathbb{C}(X) = \int_0^{+\infty} w^+(\mathbb{P}(u^+(X) > z))dz - \int_0^{+\infty} w^-(\mathbb{P}(u^-(X) > z))dz \,, \tag{1}$$

where appropriate integrability assumptions are assumed. While the CPT value $\mathbb{C}(X)$ accounts for the human agent's distortions in perception, it also recovers the expectation $\mathbb{E}(X)$ with weight functions $w^+, w^-$ and utility functions $u^+$ (resp. $-u^-$) restricted to $\mathbb{R}_+$ (resp. $\mathbb{R}_-$) are set to be the identity functions. In addition, several risk measures are also particular cases of CPT values: Variance, Conditional Value at Risk (CVar), distortion risk measures to name a few. See appendix E for proofs of these facts and Table 1 therein for a synthetic view of the settings captured by CPT.

**Problem formulation: CPT-RL.** In this work, we will focus on the policy optimization problem where the objective is the CPT value of the random variable $X = \sum_{t=0}^{H-1} r_t$ recording the cumulative rewards induced by the MDP and the policy $\pi$ for the finite horizon $H \geq 1$:

$$\max_{\pi \in \Pi_{NM}} \mathbb{C} \left[ \sum_{t=0}^{H-1} r_t \right] \,. \tag{CPT-PO}$$

We will also be concerned with the particular case of (CPT-PO) in which $w^+, w^-$ are set to the identity, namely the expected utility objective where only returns are distorted by the utility function:

$$\max_{\pi \in \Pi_{NM}} \mathbb{E} \left[ \mathcal{U} \left( \sum_{t=0}^{H-1} r_t \right) \right] \,. \tag{EUT-PO}$$

Similar problem variants for total cost and infinite horizon discounted settings can also be formulated. Notice that standard RL policy optimization problems and their risk-sensitive variants are clearly particular cases of (CPT-PO).

**Example: Personalized Treatment for Pain Management.** We illustrate our problem formulation with a concrete example in healthcare to give the reader more intuition about the different features of CPT-RL, its importance in applications when human perception and behavior matter and its differences compared to risk-sensitive RL. The goal is to help a physician manage a patient's chronic pain by suggesting a personalized treatment plan over time. The challenge here is to balance pain relief and the risk of opioid dependency or other side effects that might be due to the treatment, i.e. short-term relief and longer term risks. We propose to train a CPT-RL agent to help the physician.

1. *Why sequential decision making?* (a) The physician needs to adjust treatment at each time step depending on the patient's reported pain level as well as the observed side effects. Note here that this is relevant to dynamic treatment regimes in general (such as for chronic diseases, see e.g. Yu et al. (2021) for a survey) in which considering delayed effects of treatments is also important (and RL does account for such effects). (b) Decisions clearly impact the patient's immediate pain relief, dependency risks in the future and their overall health condition.

2. *Why CPT-RL?* Patients and clinicians make decisions influenced by psychological biases. We illustrate the importance of each one of the three features of CPT as introduced in our paper in section 2 (reference point, utility and probability distortion weight functions) via this example:

   (a) *Reference points:* Patients assess and report pain levels according to their subjective (psychologically biased) baseline. Incorporating reference point dependence leads to a more

realistic model of human decision-making taking into account *perceived* gains and losses. In our example, reducing pain from a level of 7 to 5 is not perceived the same way if the reference point of the patient is 3 of it is 5. In contrast, risk-sensitive RL treats every pain reduction as a uniform gain, regardless of the patient's starting reference pain level.

(b) *Utility transformation:* Patients might often show a loss averse behavior, i.e., they might perceive pain increase or withdrawal symptoms as worse than equivalent gains in pain relief. Note here that loss aversion should not be confused with risk aversion. In short, loss aversion can be defined as a *cognitive bias* in which the emotional impact of a loss is more intense than the satisfaction derived from an equivalent gain. For instance, in our example, a 2-point increase in pain might be seen as much worse than a 2-point reduction even if the change is the same in absolute value. This loss aversion concept is a cornerstone of Kahneman and Tversky's theory. In contrast, risk aversion rather refers to the *rational* behavior of undervaluing an uncertain outcome compared to its expected value. Risk sensitive approaches might be less adaptive to a patient's subjective preferences if they deviate from objective risk assessments.

(c) *Probability weighting:* Low probability events such as severe side effects (e.g., opioid overdose or dependency) might be overweighted or underweighted based on the patient's psychology.

**Challenges.** To conclude this section, we describe the challenges we face in solving CPT-PO. First, the CPT value does not satisfy a Bellman equation due to the nonlinearity of the utility and weight functions which breaks the additivity and linearity of the standard expected return. Second, CPT-PO is a nonconvex problem involving several nonconvex functions: The utility itself is nonconvex in general (recall the utility is convex w.r.t. gains and concave w.r.t. losses) and the probabilities are also distorted by a nonconvex weight function. While the standard policy optimization problem is already nonconvex in the policy, CPT-PO further introduces additional nonconvexity.

## 3 ABOUT OPTIMAL POLICIES IN CPT POLICY OPTIMIZATION

In this section, we investigate the properties of optimal policies to (CPT-PO) when they exist. We focus on constrasting our results with existing known results for solving standard MDPs to highlight the peculiarities of our CPT-RL problem. Understanding the properties of optimal policies are important in view of designing efficient policy search algorithms.

We start our discussion by pointing out a stark difference between optimal policies in standard MDPs and (CPT-PO). While there exists an optimal *deterministic* stationary policy for MDPs (see e.g. Thm. 6.2.10 in Puterman (2014)), this is not the case in general for (CPT-PO).

**Proposition 2.** *There does not always exist an optimal policy for* (CPT-PO) *in* $\Pi_{NM}^D$ *(i.e. deterministic non-Markovian).*

Proposition 2 tells us that the stochasticity of the policy is essential in solving our CPT-RL problem. The proof of this result is deferred to Appendix F.2: we construct a simple problem instance where an optimal policy needs to be stochastic as any deterministic policy is necessarily and clearly suboptimal. Our example is built around a $w^+$ function that puts special emphasis on the $10\%$ of the best outcomes. As a consequence, the optimal policy needs to be randomized to take advantage of this and obtain the highest returns with some probability without suffering from bad outcomes by deterministically committing to this riskier strategy. It has been briefly mentioned in L.A. et al. (2016) that the policy needs to be random in general for (CPT-PO), see also the organ transplant example in Lin et al. (2018).

The next result shows that the need for stochasticity in the optimal policy is clearly due to the probability distortions in the definition of the CPT value. Indeed, when setting the probability weight distortion function $w$ to the identity, i.e. when considering the particular case (EUT-PO) of (CPT-PO), it appears that an optimal policy is not necessarily stochastic.

**Proposition 3.** *There exists an optimal policy for* (EUT-PO) *in* $\Pi_{\Sigma,NS}^D$.

Proposition 3 allows to safely restrict our policy search to $\Pi_{\Sigma,NS}$ which is a much smaller policy space than the set of non-Markovian policies $\Pi_{NM}$. The fact that an optimal deterministic policy exists is a fundamental difference with the general (CPT-PO) setting. Whether there are specific weight functions (apart from the identity) for which there always exist a deterministic optimal policy remains an open question that we leave for future work. Proposition 3 also shows that (EUT-PO) is simpler than the more general single-trial RL problem (Mutti et al., 2023a) in which one needs to look for an optimal policy in a much larger policy set $\Pi_{NM}^D$ than $\Pi_{\Sigma,NS}^D$ in general. See appendix E.4 for the connection between both problems.

We now ask the next natural question: Can we further restrict our policy search to a smaller policy class compared to $\Pi_{\Sigma,NS}$ ? In particular, are there specific utility functions for which the resulting (EUT-PO) problem has optimal *Markovian* policies? We provide a positive answer by establishing a precise characterization of such utility functions which turn out to be either affine or exponential.

---

**Theorem 4.** *Let $\mathcal{U}$ be continuous and increasing. The following statements are equivalent:*

1. *For any MDP, there exists an optimal policy for (EUT-PO) in $\Pi_{M,NS}$.*
2. *There exists a function $\varphi : \mathbb{R}^2 \to \mathbb{R}$ such that:*
$$\forall x, a, b \in \mathbb{R}, b \neq 0, \mathcal{U}(x+a) - \mathcal{U}(x) = \varphi(a,b)(\mathcal{U}(x+b) - \mathcal{U}(x)).$$
3. *There exists a function $\mu : \mathbb{R}^2 \to \mathbb{R}$ such that:*
$$\forall y, c, d \in \mathbb{R}, \mathcal{U}(y+c) - \mathcal{U}(c) = \mu(c,d)(\mathcal{U}(y+d) - \mathcal{U}(d)).$$
4. *There exist $A, B, C \in \mathbb{R}$ s.t. $\mathcal{U}(x) = Ax + B$ or $\mathcal{U}(x) = A + B \exp(Cx)$ for all $x \in \mathbb{R}$.*

---

A few comments are in order regarding Theorem 4:

- So far, we have highlighted the importance of the probability distortion function in determining the nature of optimal policies for (CPT-PO). Theorem 4 is rather concerned with the role of the (nonlinear) utility functions in (CPT-PO).

- The theorem is reminiscent of the following known folklore result: The only memoryless continuous probability distribution is the exponential distribution.

- Theorem 4 shows that the only utility functions leading to optimal *Markovian* policies are the affine and exponential utilities. The affine utility makes (CPT-PO) boil down to a standard RL problem whereas the exponential criterion is a well-known objective used in the risk-sensitive control and RL literatures (see section 6 and appendix B for a discussion).

Theorem 4 is concerned with the (EUT-PO) problem which is a particular case of (CPT-PO). However, these results cannot be extended to (CPT-PO) in general as we show next.

---

**Proposition 5.** *There exist instances of* (CPT-PO) *where $\mathcal{U}$ is of the form $x \mapsto A + B \exp(Cx)$ for positive constants $A, B, C$ and* (CPT-PO) *does not admit an optimal policy in $\Pi_{M,NS}$.*

---

## 4 POLICY GRADIENT ALGORITHM FOR CPT-VALUE MAXIMIZATION

In this section, we propose a policy gradient algorithm for solving (CPT-PO). From this section on, we parametrize policies $\pi \in \Pi_{NM}$ by a vector $\theta \in \mathbb{R}^d$ and we denote by $\pi_\theta$ the parametrized policy. As a consequence, the CPT objective in (CPT-PO) becomes a function of the policy parameter $\theta$ and we use the shorthand notation $J(\theta)$ for the corresponding CPT objective value.

**Policy Gradient Theorem for CPT-RL.** Our key result enabling our algorithm design is a PG theorem for CPT value maximization.

**Theorem 6.** *Suppose that the utility functions $u^-, u^+$ are continuous and that the weight functions $w_-, w_+$ are Lipschitz and differentiable. Assume in addition that the policy parametrization $\theta \mapsto \pi_\theta(a|h)$ (for any $h, a \in \mathcal{H} \times \mathcal{A}$) are both differentiable. Then, for every $\theta \in \mathbb{R}^d$, the gradient of the* (CPT-PO) *objective $J$ w.r.t. the policy parameter $\theta$ is given by:*

$$\nabla J(\theta) = \mathbb{E}\left[\varphi\left(R(\tau)\right) \sum_{t=0}^{H-1} \nabla_\theta \log \pi_\theta(a_t|h_t)\right],$$

*where $\varphi(v) := \int_{z=0}^{\max(v,0)} w'_+(\mathbb{P}(u^+(R(\tau)) > z))dz - \int_{z=0}^{\max(-v,0)} w'_-(\mathbb{P}(u^-((R(\tau)) > z))dz, \forall v \in \mathbb{R}, w'_+, w'_-$ denoting the derivatives and $R(\tau) := \sum_{t=0}^{H-1} r_t$ with $\tau := (s_t, a_t, r_t)_{0 \le t \le H-1}$ is a trajectory of length $H$ generated from the MDP by following policy $\pi_\theta$.[a]*

---

[a] The integral $\varphi(R(\tau))$ is finite under our continuity assumptions since the return $R(\tau)$ is bounded.

We provide a few comments regarding this result. Theorem 6 recovers the celebrated policy gradient theorem for standard RL (Sutton et al., 1999) by setting $w_+$ (resp. $w_-$) to the identity function (on $\mathbb{R}_+$ (resp. $\mathbb{R}^-$) in which case $w'_+$ is the constant function equal to 1 and hence $\varphi(R(\tau)) = R(\tau)$. We stated the theorem in the general setting where the policy is non-Markovian. In practice, it is also possible to use a parametrization of a smaller policy set such as $\Pi_{\Sigma,NS}$ or even $\Pi_{M,S}$ in which the policy is a function of $(t, s_t, \sum_{k=0}^{t-1} r_k)$ or only $s_t$ respectively.

**Stochastic Policy Gradient Algorithm for CPT-RL.** In the light of Theorem 6, we will perform a policy gradient ascent on the objective $J$ to solve (CPT-PO). Our general policy gradient algorithm is presented in Algorithm 1. As usual, since we only have access to sampled trajectories from the MDP, we need a stochastic policy gradient to estimate the true unknown gradient given by the theorem. In particular, we need an approximation of $\varphi(R(\tau))$ for any sampled trajectory $\tau$ from the MDP following policy $\pi_\theta$. In the particular case of (EUT-PO) in which $w$ is the identity, the unknown quantity $\varphi(R(\tau))$ reduces to $\mathcal{U}(R(\tau))$ which can be easily computed as $\mathcal{U}$ is known and $R(\tau)$ is the cumulative reward.

---

**Algorithm 1** CPT-Policy Gradient Algorithm (CPT-PG) for (CPT-PO)

1: **Input:** $\theta_0 \in \mathbb{R}^d$, utility functions $u^+, u^-$, weight functions $w_+, w_-$, step size $\alpha > 0$.
2: **for** $k = 0, \cdots, K$, **do**
   /Policy gradient estimation
3:    Sample a trajectory $\tau := (s_t, a_t, r_t)_{0 \le t \le H-1}$, with $s_0 \sim \rho$ following $\pi_{\theta_k}$
   // Quantile estimation
4:    Sample $n$ trajectories $\tau_j := (s_t^j, a_t^j, r_t^j)_{0 \le t \le H-1}, 1 \le j \le n$ with $s_0^j \sim \rho$ following $\pi_{\theta_k}$
5:    Compute and order $R(\tau_j)$, label them as $R(\tau_{[1]}) < R(\tau_{[2]} < \cdots < R(\tau_{[n]})$
6:    $\hat{\xi}_{\frac{i}{n}}^+ = u^+(R(\tau_{[i]}));\ \hat{\xi}_{\frac{i}{n}}^- = u^-(R(\tau_{[i]}))$
   //Approximation of $\phi(R(\tau))$
7:    $\hat{\phi}_n^+ = \sum_{i=0}^{j_n-1} w'_+\left(\frac{i}{n}\right)\left(\hat{\xi}_{\frac{n-i}{n}}^+ - \hat{\xi}_{\frac{n-i-1}{n}}^+\right) + w'_+\left(\frac{j_n}{n}\right)\left(R(\tau) - \hat{\xi}_{\frac{n-j_n-1}{n}}^+\right)$
8:    $\hat{\phi}_n^- = \sum_{i=0}^{j_n-1} w'_-\left(\frac{i}{n}\right)\left(\hat{\xi}_{\frac{n-i}{n}}^- - \hat{\xi}_{\frac{n-i-1}{n}}^-\right) + w'_-\left(\frac{j_n}{n}\right)\left(R(\tau) - \hat{\xi}_{\frac{n-j_n-1}{n}}^-\right)$
9:    $\hat{g}_k = (\hat{\phi}_n^+ - \hat{\phi}_n^-) \sum_{t=0}^{H-1} \nabla_\theta \log \pi_{\theta_k}(a_t|h_t, \Sigma_{k=0}^{t-1} r_k, t)$
   /Policy gradient update
10:    $\theta_{k+1} = \theta_k + \alpha\, \hat{g}_k$
11: **end for**

---

In the more general setting, the approximation task becomes more challenging since we need to compute the integral term $\int w'_+(\mathbb{P}(u^+(R(\tau)) > z)dz$ (and likewise for the second integral term). We address this challenge using the following result which is a slight variation of Proposition 6 in L.A. et al. (2016) in which the integral is taken over a bounded interval. Accordingly, we end up with a different approximation formula which is tailored to the present setting. Intuitively, the approximation is a Riemann scheme approximation of the integral using simple staircase functions.

**Proposition 7.** *Let $X$ be a real-valued random variable. Suppose that the functions $w'_+, w'_-$ are Lipschitz and that $u^+(X), u^-(X)$ have bounded first moments. Let $\xi^+_{\frac{i}{n}}$ and $\xi^-_{\frac{i}{n}}$ denote the $\frac{i}{n}$th quantile of $u^+(X)$ and $u^-(X)$, respectively. Then, we have for any $v \geq 0$,*

$$\int_0^v w'_+(\mathbb{P}(u^+(X) > z))dz = \lim_{n \to \infty} \sum_{i=0}^{j_n - 1} w'_+\left(\frac{i}{n}\right)\left(\hat{\xi}^+_{\frac{n-i}{n}} - \hat{\xi}^+_{\frac{n-i-1}{n}}\right) + w'_+\left(\frac{j_n}{n}\right)\left(v - \hat{\xi}^+_{\frac{n-j_n-1}{n}}\right)$$

(2)

*where $j_n \in [0, n-1]$ is s.t. $v \in [\xi^+_{\frac{n-j_n-1}{n}}, \xi^+_{\frac{n-j_n}{n}}]$. The same identity holds when replacing $u^+(X), \xi^+_\alpha, w_+$ by $u^-(X), \xi^-_\alpha, w_-$ where $\xi^-_\alpha$ is the $\alpha^{th}$ quantile of $u^-(X)$.*

While L.A. et al. (2016) use this result to approximate the CPT value, we intend to use it for approximating our special integral terms involving the derivatives of the weight functions as they appear in the policy gradient. Using Proposition 7, we approximate the integral using a finite sum with a given number of samples $n$. As for the quantiles $\xi^+_{\frac{i}{n}}$ we compute them using the standard order statistics procedure also used in L.A. et al. (2016). Similarly to (L.A. et al., 2016, Theorem 1), our algorithm can be shown to enjoy a similar asymptotic convergence result to the set of stationary points of the (CPT-PO) objective. This is because we can also employ the same stochastic approximation artillery upon noticing that we are also approximating the same policy gradient differently and the induced bias in our case will also vanish with a large enough number of trajectories $n$ (by Thm. 6 and Prop. 16). Notice that we can also remove the projection therein upon assuming that the rewards and the score function in the policy gradient are both bounded. These fairly standard assumptions in the analysis of vanilla PG methods guarantee that the policy gradient will remain bounded.

**Comparison to the CPT-SPSA-G algorithm in L.A. et al. (2016).** Our algorithm is specifically designed for maximizing the CPT value of a (discounted) sum of rewards generated by an MDP while the CPT-SPSA-G algorithm in L.A. et al. (2016) can be used for a larger class of problems to maximize the CPT value of any real-valued random variable. However, we highlight that (a) this cumulative reward return structure is natural and ubiquitous in RL and economics applications and foremost (b) thanks to this particular problem structure, our algorithm is a policy gradient algorithm leveraging first-order information whereas CPT-SPSA-G only uses zeroth order information, i.e. CPT value estimations. This difference is crucial as zeroth order optimization algorithms are known to suffer from the curse of dimensionality. Our algorithm can scale better to higher dimensional problems as it is notoriously known for policy gradient algorithms in classic RL. We provide empirical evidence of this fact in section 5 to further support the benefits of our algorithm.

## 5 EXPERIMENTS

We demonstrate the performance of our CPT-PG algorithm in three different settings: (a) we consider a traffic control application to show the influence of the probability distortion function, (b) we illustrate the better scalability of our PG algorithm to larger state spaces compared to the existing zeroth order algorithm in a grid environment with increasing state space size and (c) we show the applicability and performance of our algorithm in a continuous state-action space setting via an electricity management application. See appendix 3 for more details (Table 2 therein) and additional simulations illustrating some of our theoretical findings of section 3 (Proposition 2 and Theorem 4).

**(a) Traffic Control.** We consider a car agent which would like to reach a given destination at the other side of the city. Passing through the city center is faster on average but carries a small risk of incurring a very large delay. We model the setting as a $n \times n$ grid (see fig. 1 center). Central roads can get cluttered and peripheral roads take constant time to get through. We run our PG algorithm, the training curves are reported in Fig. 2 (center). In the risk-neutral case, we observe that the total expected return is higher than in the CPT case. This is because the risk averse policy compromises return in order to get certainty by going around the risky city center. These examples show that our algorithm is successful at finding different optimal strategies for different weight functions $w$.

**Influence of the utility function.** We consider a 4x4 grid for our illustration purpose. Our agent starts on a random square on one of the three upper rows of the grid, and can move in all four directions. Any move to an empty square will award it a random reward of $-1$ with probability $\frac{1}{2}$ and of

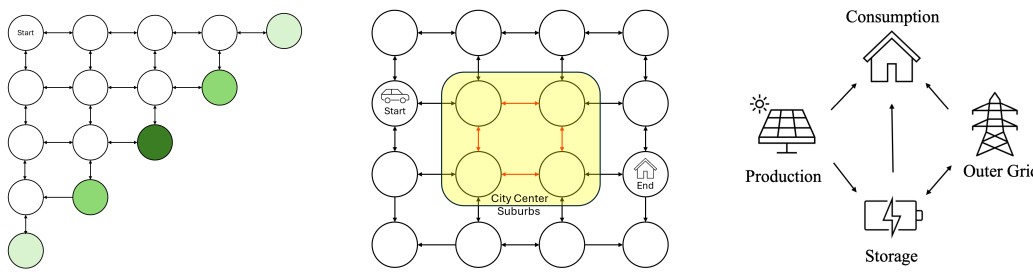

Figure 1: (**Left**) Scaling grid example. (**Center**) Traffic control: red roads in the city center are prone to congestion. (**Right**) Electricity management: Arrows refer to electricity flow.

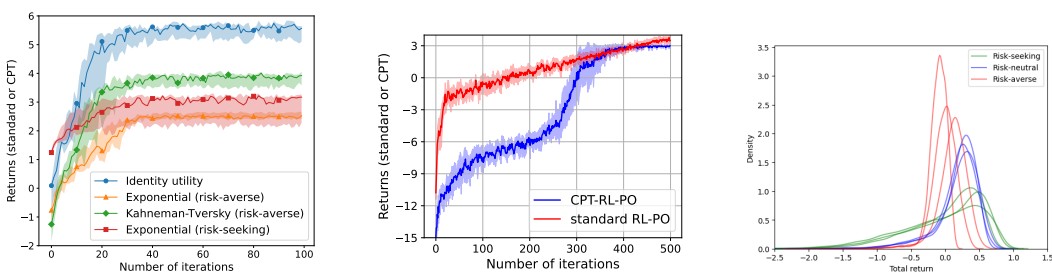

Figure 2: Returns along the iterations of our PG algorithm for (CPT-PO) for: (**Left**) different utility functions with the same distortion $w$ in the grid environment, (**center**) traffic control. Shaded areas indicate a range of $\pm$ one standard deviation over 20 runs. (**Right**) Density of the empirical returns obtained by deploying different trained PG policies (from different initializations) for electricity management, density is computed using 10000 runs for each curve. See appendix H for details.

$+0.8$ with probability $\frac{1}{2}$. Therefore, longer trajectories are slightly costly in expectation, and generate significant variance. In two corners of the grid, we add cells that yields rewards of $+5$ for one or $+6$ for the other, and conclude the episode. Illegal moves (attempting to leave the grid) are punished by a negative reward. Our parameterized policy is a neural network whose last layer is activated with softmax and has 4 coordinates corresponding to the 4 different possible moving actions. We consider solving (CPT-PO) with different utility functions: risk-neutral identity utility, risk-averse KT utility, as well as exponential utility function. The obtained policies differ depending on the utility function. For examples of risk-neutral/averse policies obtained, see Fig. 16b in appendix H.4.

**(b) Scalability to larger state spaces.** We now compare our PG algorithm to the zeroth-order algorithm of L.A. et al. (2016) (CPT-SPSA-G). We consider a family of MDPs where the state space is a $n \times n$ grid for a given integer parameter $n$. The agent starts in the top right corner and has always four possible actions (up,down,left,right). Taking a step yields a reward of $\frac{-1}{n}$, attempting to leave the grid yields $\frac{-2}{n}$, and reaching the anti-diagonal ends the episode with a positive reward. All cells on the anti-diagonal yield the same expected reward, but with different levels of risk; the least risky reward is the deterministic one, in the center of the grid. We consider tabular policies and the initial policy is a random policy assigning the probability $1/4$ to each action. We test the sensitivity of the performance of both algorithms to the size of the state space. The steps sizes of both algorithms have been tuned through trial and error in an effort to approach their possible performance; we wish to draw attention not to the absolute performance of either algorithm on any particular example, but rather to the evolution of the performance of both as the size of the problem increases. We observe that the performance of CPT-SPSA-G suffers for larger state space size whereas our PG algorithm is robust to state space scaling. While both algorithms are gradient ascent based algorithms in principle, our stochastic policy gradients are different.

**(c) Electricity management.** Our goal now is to show the performance of our algorithm in a *continuous* state and action space setting. We consider an electricity management system for an individual

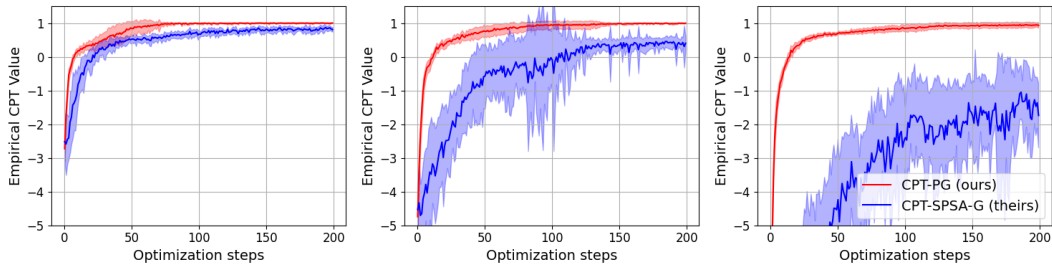

Figure 3: Compared performance of our algorithm and CPT-SPSA-G for $n = 3, 5, 9$. The shaded area is a range of $\pm$ one standard deviation over 10 independent runs.

home which has solar panels for producing electricity and a battery. The intensity of the solar panel's electricity production follows a sinusoidal function during daytime hours and vanishes at night. The home consumes a random and varying amount of electricity and can buy and sell electricity to the outer grid. The selling price varies during the day whereas the buying price is fixed and significantly higher than the selling prices. We use public data for selling prices recorded on the French electricity network (see appendix for further details). We consider a 24h time frame starting at 6 am and we divide the day into twelve two-hour time slots. For each time slot, the agent has to decide how much to buy or sell to the grid given the production, the battery's charge, the price on the market and the consumption. We run the algorithm with three different objectives, changing the $w$ function: a risk-neutral one, a risk-averse one and a risk-seeking one (see Table 2). We consider a Gaussian policy in which the mean is parameterized by a neural network. We report the results for running our algorithm in Fig. (2) (right). The most rewarding time to sell our electricity is around 4pm (see electricity prices in appendix H.6, Fig. 19, right). However, selling too much too soon exposes us to the risk of falling short of battery during the night and risking to buy it later for a higher price.

The risk-averse policy avoids selling a lot of electricity and tends to keep it stored until the end of the day. Conversely, the risk-seeking policy aggressively sells energy when the markets are high at the cost of possibly having to buy it again later in the day. We can see on Fig. 2 (right), where we plot the distribution of total returns for various trained models with different $w$ functions and a few different random initializations each, that, as we would wish to see, the risk-averse policy has the distribution with the best left tail (worst cases are not too bad), the risk-seeking distribution has the best right tail (best cases are particularly good). The risk-neutral policy has the best mean value.

## 6 RELATED WORK

We refer the reader to appendix B for an extended related work discussion including CPT-RL, convex RL and risk-sensitive RL. See also appendix E.1 for a diagram relating them.

## 7 CONCLUSION

We investigated a CPT variant of the standard RL problem to model human decision making. We provided new insights on optimal policies in such problems to highlight their peculiarity compared to classical RL. Then, we designed a novel PG algorithm for CPT-PO. Finally, we showed the benefits of our algorithm in terms of scalability compared to prior work and we illustrated its performance in applications including electricity management and traffic control. Our work opens the way to interesting avenues for future work. Using CPT usually requires to know the utility and distortion functions (or to posit models thereof) a priori. Can we learn such functions to align with the preferences of the human decision maker involved? Looking forward, investigating incentive design problems in which human agents are collectively modelled using CPT would be interesting. We hope our work will stimulate further research in better capturing the behavior and preferences of human agents in real-world decision making applications beyond expected utility.

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

CONTENTS

# A   NOTATION FOR POLICY CLASSES

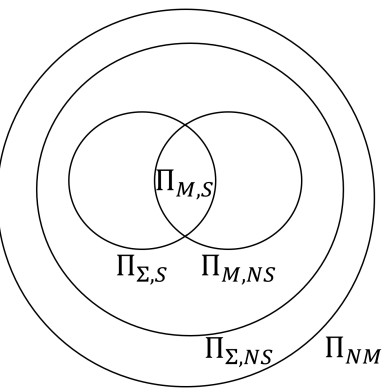

Figure 4: Policy classes (see Rem. 1).

Throughout this work, we will consider the following sets of policies:

- $\Pi_{NM} := \{\mathcal{H} \to \Delta(\mathcal{A})\}$ is the set of non-Markovian policies,[2]

- $\Pi_{\Sigma,NS} := \{\mathcal{S} \times \mathbb{R} \times \mathbb{N} \to \Delta(\mathcal{A})\}$ is the set of policies that only depend on the current state, the timestep and the sum of discounted rewards accumulated so far: The RL agent in state $s$ at timestep $t$ following policy $\pi \in \Pi_{\Sigma,NS}$ samples its next action from the distribution $\pi(s, \sum_{k=0}^{t-1} \gamma^k r_k, t)$,

- $\Pi_{\Sigma,S} := \{\mathcal{S} \times \mathbb{R} \to \Delta(\mathcal{A})\}$ is the set of policies that only depend on the state and the sum of discounted rewards: The RL agent in state $s$ at timestep $t$ following policy $\pi \in \Pi_{\Sigma,S}$ samples its next action from the distribution $\pi(s, \sum_{k=0}^{t-1} \gamma^k r_k)$,

- $\Pi_{M,NS} := \{\mathcal{S} \times \mathbb{N} \to \Delta(\mathcal{A})\}$ is the set of Markovian policies: An agent in state $s$ at timestep $t$ following policy $\pi \in \Pi_{M,NS}$ samples its next action from the distribution $\pi(s, t)$.

- $\Pi_{M,S} := \{\mathcal{S} \to \Delta(\mathcal{A})\}$ is the set of stationary Markovian policies, i.e. Markovian policies which are time-independent.

# B   EXTENDED RELATED WORK DISCUSSION

**Risk-sensitive RL.** There is a rich literature around risk sensitive control and RL that we do not hope to give justice to here. We refer the reader to recent comprehensive surveys on the topic (García & Fernández, 2015; Prashanth et al., 2022) and the references therein. Let us briefly mention that there exist several approaches to risk sensitive RL. These include formulations such as constrained stochastic optimization to control the tolerance to perturbations and stochastic minmax optimization to model robustness with respect to worst case perturbations for instance. Another approach which is more relevant to our paper discussion consists in regularizing or modifying objective functions.

---

[2]By 'non-Markovian', we mean '*non necessarily* Markovian' policies including Markovian ones. Elements of $\Pi_{NM} - \Pi_{M,NS}$ can be designated as 'stricly non-Markovian' policies. Likewise, by 'non stationary', we mean 'non necessarily stationary', and by 'stochastic' we mean 'non necessarily deterministic'.

Such modifications are based on considering different statistics of the return deviating from the standard expectation such as the variance or the conditional value at risk (e.g. Tamar et al. (2012); Chow & Ghavamzadeh (2014); Chow et al. (2018)) or even considering the entire distribution of the returns like distributional RL (Bellemare et al., 2023). Another popular objective modification consists in maximizing an exponential criterion (e.g. Borkar (2002); Noorani et al. (2022)) to obtain robust policies w.r.t noise and perturbations of system parameters or variations in the environment. Noorani et al. (2022) designed a model-free REINFORCE algorithm and an actor-critic variant of the algorithm leveraging an (approximate) multiplicative Bellman equation induced by the exponential objective criterion. Moharrami et al. (2024) recently proposed and analyzed similar PG algorithms for the same exponential objective. Vijayan & LA (2023) introduced a PG algorithm for solving risk-sensitive RL for a class of smooth risk measures including some distortion risk measures and a mean-variance risk measure. Their approach is based on simultaneous perturbation stochastic approximation (SPSA) (Bhatnagar et al., 2013) using zeroth-order information to estimate gradients. Our CPT-PO problem covers several of the aforementioned objectives including smooth distortion risk measures and exponential utility as particular cases (see appendix E for more details).

**Convex RL/RL with General Utilities.** In the last few years, convex RL (a.k.a. RL with general utilities) (Hazan et al., 2019; Zhang et al., 2020; Zahavy et al., 2021; Geist et al., 2022) has emerged as a framework to unify several problems of interest such as pure exploration, imitation learning or experiment design. More precisely, this line of research is concerned with maximizing a given functional of the state(-action) occupancy measure w.r.t. a policy. To solve this problem, several policy gradient algorithms have been proposed in the literature (Zhang et al., 2021; Barakat et al., 2023). Mutti et al. (2022b;a; 2023a) challenged the initial problem formulation and proposed a finite trial version of the problem which is closer to practical concerns as it consists in maximizing a functional of the empirical state(-action) distribution rather than its true asymptotic counterpart. The particular case of our CPT policy optimization problem without probability distortion (see (EUT-PO) below) coincides with a particular case of the single trial convex RL problem (Mutti et al., 2023b) in which the function of the empirical visitation measure is a linear functional of the reward function (see appendix E.4 for details). However, our general problem is not a particular case of convex RL which does not account for probability distortions. Furthermore, our utility function is in general non-convex in our setting (see example in Fig 6) and our policy gradient algorithm is model-free. More recently, De Santi et al. (2024) introduced a *global* RL problem formulation where rewards are globally defined over trajectories instead of locally over states and used submodular optimization tools to solve the resulting non-additive policy optimization problem. While global RL allows to account for trajectory-level global rewards, it does not take into consideration probability distortions. In addition, their investigation is restricted to the setting where the transition model is known whereas our PG algorithm is model-free.

**Cumulative Prospect Theoretic RL.** Motivated by Prospect Theory and its sibling CPT (Kahneman & Tversky, 1979; Tversky & Kahneman, 1992; Barberis, 2013), L.A. et al. (2016) first proposed to combine CPT with RL to obtain a better model for human decision making. Following this first research effort, only few isolated works (Borkar & Chandak, 2021; Ramasubramanian et al., 2021; Ethayarajh et al., 2024) considered a similar CPT-RL setting. In particular, Borkar & Chandak (2021) proposed and analyzed a Q-learning algorithm for CPT policy optimization. Ramasubramanian et al. (2021) further developed value-based algorithms for CPT-RL by estimating the CPT value of an action in a given state via dynamic programming. More precisely, they were concerned with maximizing a sum of CPT value period costs which is amenable to dynamic programming. In contrast to their accumulated CPT-based cost (see their remark 1), our CPT policy optimization problem formulation is different: we maximize the CPT value of the return of a policy (see (CPT-PO)). In particular, this objective does not enjoy an additive structure and hence does not satisfy a Bellman equation. Moreover, their work relying on value-based methods is restricted to finite discrete state action spaces. Our PG algorithm is also suitable for continuous state action settings as we demonstrate in our experiments. More recently, Ethayarajh et al. (2024) incorporated CPT (without probability distortion) into RL from human feedback for fine-tuning large language models. CPT has also been recently exploited for multi-agent RL (Danis et al., 2023). Our work is complementary to this line of research, especially to L.A. et al. (2016) and its extended version Jie et al. (2018) which are the most closely related work to ours. While their algorithm design makes use of simultaneous perturbation stochastic approximation (SPSA) (Spall, 1992) using only zeroth order information, we rather propose a PG algorithm exploiting first-order information thanks to our

special problem structure involving the CPT value of a cumulative sum of rewards. See section 4 for further details regarding this comparison.

We refer the reader to Appendix E.1 for a summarizing diagram illustrating the relationships between CPT-RL, convex RL and risk-sensitive RL.

## C    APPLICATIONS OF CPT

In this section, we provide a discussion regarding the applications where CPT has already been successfully used (mainly in the static stateless setting) and potential applications in the dynamic (RL) setting with state transitions.

We highlight that CPT has been tested and effectively used in a large number of compelling behavioral studies that we cannot hope to give justice to here. Besides the initial findings of Tversky and Kahneman for which the latter won the Nobel Prize in economics in 2002, please see a few recent references below for a broad spectrum of real-world applications ranging from economics to transport, security and energy, mostly in the stateless (static) setting.

- Risk preferences across 53 countries worldwide in an international survey (Rieger et al., 2017). Estimates of CPT parameters from data illustrate economic and cultural differences whereas probability weighting also reflects gender differences as well as economic and cultural impacts. Note here the explainability feature of CPT.
- A study of homeowners in the Netherlands to investigate energy retrofit decision using CPT (Ebrahimigharehbaghi et al., 2022). CPT is shown to predict the number of homeowners decisions to renovate their homes more accurately than Expected Utility Theory (EUT).
- Application of CPT to building evacuation (Gao et al., 2023). CPT allows to take into account individual psychology and irrational behavior in modeling evacuations via pedestrian movement modeling. This is particularly important for designing and optimizing emergency and safety management strategies.
- Understanding private parking space owners' propensity to share their parking spaces by considering their psychological concerns as well as their socio-demographic and revenue characteristics for instance (Yan et al., 2020). This might be useful to help developing shared parking services.
- Home energy management (Dorahaki et al., 2022). This work proposes a behavioral home energy management model to increase the user's satisfaction.
- Empirical study about financial decision making in two universities in Argentina (Ladrón de Guevara Cortés et al., 2023). In particular, it is shown that the financial decisions of the participants under uncertainty are more consistent with Prospect Theory than expected utility theory.

Our CPT-RL problem formulation finds applicatons in a number of diverse areas. A nonexhaustive list includes:

- **Traffic control.** We refer the reader to our toy example in the main part. simulations for specific CPT-RL applications in simple settings for traffic control, electricity management and financial trading that we will not discuss again here.
- **Electricity management.** Please see simulations in the main part (section 5 and appendix H.6) in a simple example setting to illustrate our methodology.
- **Finance:** portfolio optimization, risk management, behavioral asset pricing (e.g. influence of investor sentiment on price dynamics via e.g. over-weighting of low-probability events, including their preferences). For recent applications of CPT to finance, we refer the reader to a recent paper Luxenberg et al. (2024) using CPT for portfolio optimization (in a stateless static setting). We also applied our methodology to financial trading (see Appendix H.7).
- **Health:** personalized treatment plans, (e.g. health insurance design for specific groups modeling risk and factoring perceived fairness).

On a more high-level note, we would like to mention that CPT-RL is of practical relevance for finance and healthcare for several reasons: in short, CPT allows for (a) **modeling human biases**, (b) **factoring risk**, and (c) **capturing individual preferences for personalization.** All these three points are essential in the above applications.

## D CPT-RL AND TRAJECTORY-BASED REWARD RL AS PREFERENCE LEARNING PARADIGMS

In this section, we compare the CPT-RL and trajectory-based reward RL (using a single reward for the entire trajectory, such as Reinforcement Learning from Human Feedback) seen as preference learning paradigms. In particular, we also discuss the pros and cons of each one of them.

Regarding the structure of the final reward and the metric learning you mention, this is a fair point and we agree that Our present work requires so far access to utility and weight functions whereas trajectory-based reward RL learns the metric to be optimized using human preference data. However, let us mention a few points:

(a) These can be readily available in specific applications (for risk modeling or even chosen at will by the users themselves);

(b) CPT relies on a predefined model, this can be beneficial in applications such as portfolio optimization or medical treatment where trade-offs have to be made and models might be readily available;

(c) Furthermore, we argue that having such a model allows it to be more explainable compared to a model entirely relying on human feedback and fine tuning, let alone the discussion about the cost of collecting human feedback. We also note that some of the most widely used algorithms in RLHF (e.g. DPO) do rely on the fact that the reward follows a Bradley-Terry model for instance (either for learning the reward or at least to derive the algorithm to bypass reward learning);

(d) Let us mention that one can also learn the utility and weight functions. We mentioned this promising possibility in our conclusion although we did not pursue this direction in this work. One can for instance represent the utility and weight functions by neural networks and train models to learn them using available data with relevant losses, jointly with the policy optimization task. One can also simply fit the predefined functions (say e.g. Tversky and Kahneman's function) to the data by estimating the parameters of these functions (see $\eta$ with our notations and exponents of the utility function in Table 1 for the CPT row). This last approach is already commonly used in practice, see e.g. Rieger et al. (2017).

**CPT vs RLHF: General comparison.** CPT has been particularly useful when modeling specific biases in decision making under risk to account for biased probability perceptions. It allows to *explicitly* model cognitive biases. In contrast, RLHF has been successful in training LLMs which are aligned with human preferences where these are complex and potentially evolving and where biases cannot be explicitly and reasonably modeled. RLHF has been rather focused on learning *implicit* human preferences through interaction (e.g. using rankings and/or pairwise comparisons). Overall, CPT can be useful for tasks where risk modeling is essential and critical whereas RLHF can be useful for general preference alignment although RLHF can also be adapted to model risk if human preferences are observable and abundantly available at a reasonable cost. This might not be the case in healthcare applications for instance, where one can be satisfied with a tunable risk model. On the other hand, so far CPT does not have this ability to adapt to evolving preferences over time unlike RLHF which can do so via feedback.

**CPT and RLHF: Pros and cons.** To summarize the pros and cons of both approaches, we provide the following elements. As for the pros, CPT directly models psychological human biases in decision making via a structured framework which is particularly effective for risk preferences. RLHF can generalize to different scenarios with sufficient feedback and handle complex preferences via learning from diverse human interactions, it is particularly useful in settings where preferences are not explicitly defined such as for LLMs for aligning the systems with human preferences and values. As for the cons, CPT is a static framework since the utility and probability weight functions are fixed, it is hence less adaptive to changing preferences. It uses a predefined model of human behavior which is not directly using feedback. It also requires to estimate model parameters precisely, often for specific domains. As for RLHF on the other hand, the quality and the quantity of the human feedback is essential and this dependence on the feedback clearly impacts performance. This dependence can also cause undesirable bias amplification which is present in the human feedback. We also note that training such models is computationally expensive in large scale applications.

**CPT and RLHF are not mutually exclusive.** While CPT and trajectory-based RL (say e.g. RLHF) both offer frameworks for incorporating human preferences into decision making, we would like to highlight that CPT and RLHF are not mutually exclusive. We can for instance use CPT to design an initial reward structure reflecting human biases, then refine it with RLHF. We can also consider to further relax the requirement of sum of rewards (which already has several applications on its own) and think about incorporating CPT features to RLHF. Some recent efforts in the literature in this direction that we mentioned in our paper include the work of Ethayarajh et al. (2024) which combines prospect theory with RLHF (without probability weight distortion though, which limits its power). Note that the ideas of utility transformation and probability weighting are not crucially dependent on the sum of rewards structure and can also be applied to trajectory-based rewards or trajectory frequencies for instance. We believe this direction deserves further research, one interesting point would be how to incorporate risk awareness from human behavior to such RLHF models using ideas from CPT.

# E   COMPLEMENTS ABOUT CPT VALUES AND CPT POLICY OPTIMIZATION

## E.1   POSITIONING CPT-RL IN THE LITERATURE

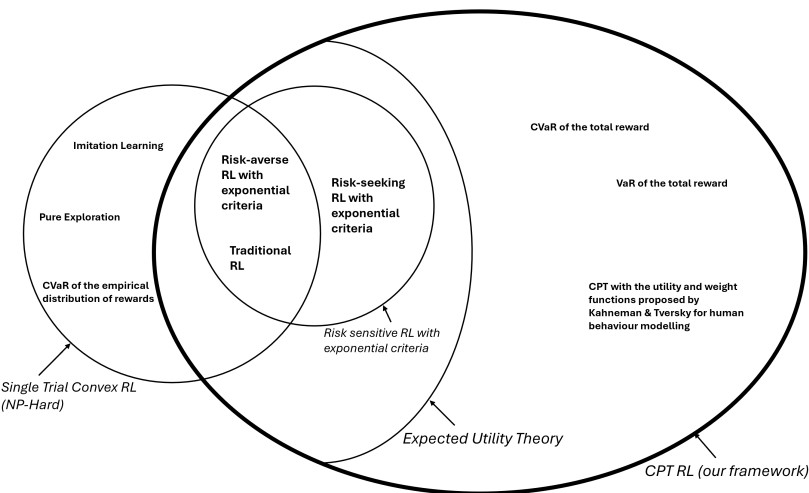

Figure 5: A Venn Diagram representing our framework and some other frameworks in the literature

**Remark 8.** *For the infinite horizon discounted setting, the objective becomes the CPT value of the random variable $X = \sum_{t=0}^{+\infty} \gamma^t r_t$ recording the cumulative discounted rewards induced by the MDP and the policy $\pi$. The policy can further be parameterized by a vector parameter $\theta \in \mathbb{R}^d$.*

## E.2   CPT VALUE EXAMPLES

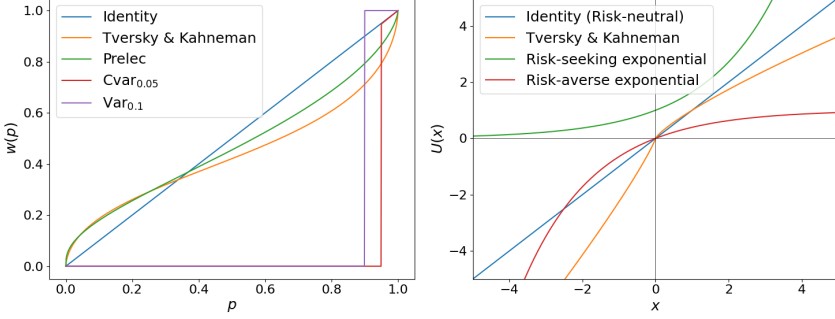

Figure 6: Various examples of probability weight functions (left) and utility functions (right).

| Setting | Utility function | $w^+$ | $w^-$ |
|---|---|---|---|
| CPT | Any | Any | Any |
| CPT (Functions proposed by Kahneman and Tversky) | $\begin{cases} (x - x_0)^\alpha & \text{if } x \geq 0, \\ -\lambda(x - x_0)^\alpha & \text{if } x < 0 \end{cases}$ | $\dfrac{p^\eta}{(p^\eta + (1-p)^\eta)^{\frac{1}{\eta}}}$ | $\dfrac{p^\delta}{(p^\delta + (1-p)^\delta)^{\frac{1}{\delta}}}$ |
| EUT | Any | Identity function | Identity function |
| Distortion risk measure | Identity function | Any | $1 - w^+(1-t)$ |
| CVaR* (Balbás et al. (2009)) | Identity function | $1 - w^-(1-t)$ | $\begin{cases} \frac{x}{1-\alpha} & \text{if } 0 \leq x < 1 - \alpha, \\ 1 & \text{if } 1 - \alpha \leq x \leq 1 \end{cases}$ |
| VaR* (Balbás et al., 2009) | Identity function | $1 - w^-(1-t)$ | $\begin{cases} 0 & \text{if } 0 \leq x < 1 - \alpha, \\ 1 & \text{if } 1 - \alpha \leq x \leq 1 \end{cases}$ |
| Risk-sensitive RL with exponential criteria (Noorani et al., 2022) | $\frac{1}{\beta} \exp(\beta x), \beta > 0$ | Identity function | Identity function |

Table 1: CPT value examples. *: $w^+$ and $w^-$ are often required to be continuous, which would exclude VaR and CVaR.

### E.3   PROOF: CVAR, VAR AND DISTORTION RISK MEASURES ARE CPT VALUES

For a random variable $X$ and a non-decreasing function $g : [0, 1] \to [0, 1]$ with $g(0) = 0$ and $g(1) = 1$, the **distortion risk measure** (Sereda et al., 2010) is defined as:

$$\rho_g(X) := \int_{-\infty}^0 \tilde{g}(F_{-X}(x))dx - \int_0^{+\infty} g(1 - F_{-X}(x))dx \,,$$

where $F_{-X} : t \mapsto \mathbb{P}(-X \leq t)$ and $\tilde{g} : t \mapsto 1 - g(1 - t)\,.$

**Proposition 9.** *Any distortion risk measure of a given random variable $X$ can be written as a CPT value with $u^+ = id^+$, $u^- = -id^-$, $w^+ = \tilde{g}$ and $w^- = g\,.$*

*Proof.* It follows from the definition of the distortion risk measure together with a simple change of variable $x \mapsto -x$ that:

$$\rho_g(X) = \int_{-\infty}^0 \tilde{g}(F_{-X}(x))dx - \int_0^{+\infty} g(1 - F_{-X}(x))dx$$

$$= -\int_{+\infty}^0 \tilde{g}(F_{-X}(-x))dx - \int_0^{+\infty} g(1 - F_{-X}(x))dx$$

$$= \int_0^{+\infty} \tilde{g}(F_{-X}(-x))dx - \int_0^{+\infty} g(1 - F_{-X}(x))dx$$

$$= \int_0^{+\infty} \tilde{g}(\mathbb{P}(-X \leq -x))dx - \int_0^{+\infty} g(1 - \mathbb{P}(-X \leq x))dx$$

$$= \int_0^{+\infty} \tilde{g}(\mathbb{P}(X \geq x))dx - \int_0^{+\infty} g(\mathbb{P}(-X > x))dx\,.$$

Since $g(\mathbb{P}(-X > x)) = g(\mathbb{P}(-X \geq x))$ almost everywhere (in a measure theoretic sense) on $[0, +\infty($, and $g$ is bounded, we obtain:

$$\rho_g(X) = \int_0^{+\infty} \tilde{g}(\mathbb{P}(X \geq x))dx - \int_0^{+\infty} g(\mathbb{P}(-X \geq x))dx\,.$$

We recognize the CPT-value of $X$ with $u^+ = \mathrm{id}^+$, $u^- = -\mathrm{id}^-$, $w^+ = \tilde{g}$ and $w^- = g\,.$ $\qquad\square$

**Remark 10.** *When $X$ admits a density function, Value at Risk (VaR) and Conditional Value at Risk (CVaR) (Wirch & Hardy (2001)) have been shown to be special cases of distortion risk measures and are therefore also instances of CPT-values.*

### E.4 CONNECTION TO GENERAL UTILITY RL AND CONVEX RL IN FINITE TRIALS

In this section, we elaborate in more details on one of the connections we noticed (and mentioned in related works) between our (CPT-PO) problem of interest and the literature of generality utility RL.

The general utility RL problem consists in maximizing a (non-linear in general) functional of the occupancy measure induced by a policy. More formally, the general utility RL can be written as follows:

$$\max_\pi F(d_\rho^\pi), \tag{3}$$

where $F$ is the real valued utility function defined on the set of probability measures over the state or state-action space, $\rho$ is the initial state distribution and $d_\rho^\pi$ is the state (or sometimes state-action) occupancy measure induced by the policy $\pi$. This problem captures the standard RL problem as a particular case by considering a linear functional $F$ defined using a fixed given reward function. Recently, motivated by practical concerns, Mutti et al. (2023b) argued for the relevance of a variation of the problem under the qualification of *convex RL in finite trials*. They introduce for this the empirical state distributions $d_n \in \Delta(\mathcal{S})$ defined for every state $s \in \mathcal{S}$ by:

$$d_n^\pi(s) = \frac{1}{nT} \sum_{i=1}^n \sum_{t=0}^{T-1} \mathbb{1}(s_{t,i} = s), \tag{4}$$

where $s_{t,i}$ is the state at time $t$ resulting from the interaction with the MDP (with policy $\pi$) in the $i$-th episode, among $n$ independent trials. Their policy optimization problem is then as follows:

$$\max_\pi \xi_n(\pi) := \mathbb{E}[F(d_n^\pi)]. \tag{5}$$

Note that $d_n^\pi$ is a random variable as it is an empirical state distribution. Observe also that $\lim_{n\to\infty} \xi_n(\pi) = F(d_\rho^\pi)$ under mild technical conditions (e.g. continuity and boundedness of $F$). This shows the connection between the above final trial convex RL objective and the general utility RL problem (3). The interesting differences between both problem formulations arise for small values of $n$. Of particular interest, both in this paper and in Mutti et al. (2023b), is the *single trial RL* setting where $n = 1$.

Setting the probability distortion function $w$ to be the identity, our (CPT-PO) problem becomes (EUT-PO), i.e.:

$$\max_\pi \mathbb{E}\left[\mathcal{U}\left(\sum_{t=0}^{H-1} r_t\right)\right], \tag{6}$$

which is of the form $\xi_1(\pi)$, the single-trial RL objective as defined in Mutti et al. (2023b). Indeed, it suffices to write the following to observe it:

$$\mathcal{U}\left(\sum_{t=0}^{H-1} r_t\right) = \mathcal{U}(\langle d_1^\pi, r \rangle), \tag{7}$$

where $r$ is the reward function seen as a vector in $\mathbb{R}^{|\mathcal{S}|}$, $\langle \cdot, \cdot \rangle$ is the standard Euclidean product in $\mathbb{R}^{|\mathcal{S}|}$. Therefore, it appears that the above objective is indeed a functional of the empirical distribution $d_1^\pi$. Single trial general utility RL is more general than (EUT-PO) since it does not necessarily consider an additive reward inside the non-linear utility and can accommodate any (convex) functional of the occupancy measure. However, (CPT-PO) does not appear to be a particular case of single trial convex RL because of the probability distortion function introduced.

## F PROOFS FOR SECTION 3

### F.1 UNWINDING MDPS FOR CPT-RL

In this section, we describe an equivalent MDP construction that will be used in some of our proofs such as for Proposition 3. For any CPT-MDP $(\mathcal{S}, \mathcal{A}, r, P)$ with utility function $\mathcal{U}$, we can formally define an equivalent 'unwinded' MDP[3] that can be solved using classical RL techniques. For any

---

[3]This terminology is not standard, we adopt it here to describe our approach.

state $s \in \mathcal{S}$ is the original MDP and any timestep $t \leq H - 1$ with cumulative reward $\sum_{k=0}^{t} r_k$, we associate a state $\tilde{s} := (s, t, \sum_{k=0}^{t} r_k)$ and the rewards in the unwinded MDP are adjusted as to reflect the difference in utility between two consecutive states:

$$\tilde{r}_t = \mathcal{U}\left(\sum_{k=1}^{t+1} r_k\right) - \mathcal{U}\left(\sum_{k=1}^{t} r_k\right). \tag{8}$$

We observe that all the information needed at any given timestep to take a decision on the next action to take is contained in $\tilde{s}$. This implies that any CPT-value that can be achieved by a non-Markovian strategy on the original MDP can also be achieved by a Markovian policy on the unwinded MDP.

The reader might notice that the size of the unwinded MDP grows with the horizon length and might blow up depending on the original MDP structure. As a consequence, learning in this unwinded MDP might become intractable. If the original MDM can be represented as a finite directed acyclic graph, the unwinded MDP is also a finite directed acyclic graph. If the underlying MDP contains a cycle, even if it is finite, its unwinded version may contain an infinite number of states. In the case of a stochastic tree MDP, the unwinded MDP has the exact same shape as the original one.

Note that we will only be using the unwinded MDP as a theoretical construction to prove some of our results and we do not perform any learning task in this unwinded MDP.

### F.2 PROOF OF PROPOSITION 2

To prove the proposition, we consider a simple MDP with only two states (an initial state and a terminal one) and two actions (A and B). See Fig. 12a below. We choose the identity as utility. Action A yields reward 1 with probability 1 and action B yields either 0 or $\frac{3}{2}$ with probability $\frac{1}{2}$ each. We further consider the following probability distortion function $w^+ : [0, 1] \to [0, 1]$ defined for every $x \in [0, 1]$ as follows:

$$w^+(x) = \begin{cases} 5x & \text{if } x \leq 0.1, \\ \frac{1}{2} + \frac{5}{9}(x - 0.1) & \text{otherwise}, \end{cases} \tag{9}$$

and we set $w^- = 0$. All the policies can be described with a single scalar $p \in [0, 1]$, the probability of choosing B instead of A.

The CPT value of the reward $X$ is:

$$\mathbb{C}(X) = w^+\left(1 - \frac{p}{2}\right) + \frac{1}{2}w^+\left(\frac{p}{2}\right). \tag{10}$$

There are only two possible deterministic policies:

- For the policy corresponding to $p = 0$, $\mathbb{C}(X) = 1$.

- For the policy corresponding to $p = 1$, $\mathbb{C}(X) = \frac{3}{2}w^+(\frac{1}{2}) = \frac{13}{12} \approx 1.08$.

However, with the non-deterministic policy $p = 0.2$, we get:

$$\mathbb{C}(X) = w^+(0.9) + \frac{1}{2}w^+(0.1) = \frac{17}{18} + \frac{1}{4} = \frac{43}{36} \approx 1.19$$

which is larger than the CPT values of both deterministic policies. We conclude that there are no deterministic policies solving the CPT problem in this case.

**Remark 11.** *We provided a counterexample with random rewards, but there also exist counterexamples with deterministic rewards. One way to build such a counterexample is to start from the MDP we just studied and 'transfer' the randomness from the reward functions to the probability transition, by constructing a larger -but equivalent- MDP, with intermediate states like in Fig. 8.*

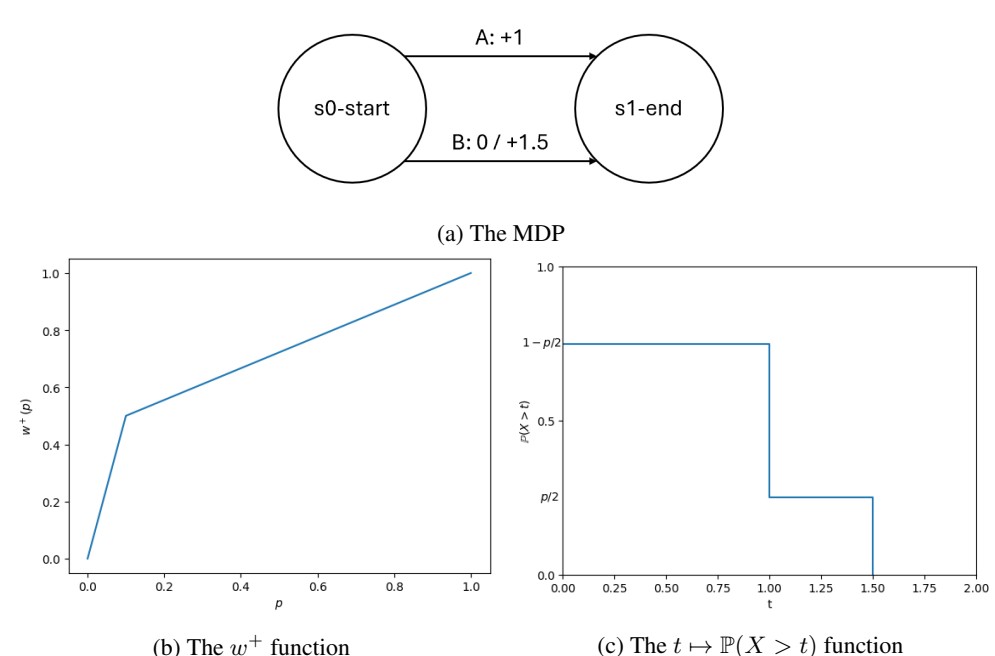

(a) The MDP

(b) The $w^+$ function

(c) The $t \mapsto \mathbb{P}(X > t)$ function

Figure 7: Problem instance for the proof of Proposition 2.

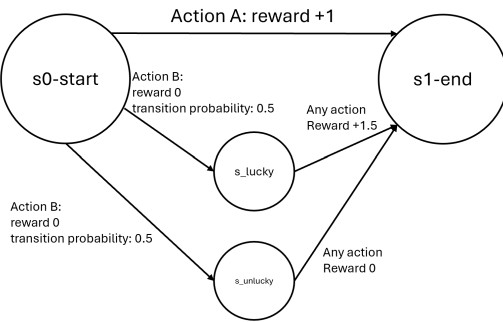

Figure 8: An equivalent example with deterministic rewards

## F.3 PROOF OF PROPOSITION 3

We build a classical MDP which is equivalent to our CPT-MDP following the procedure described in more details in appendix F.1. In summary, we expand every state $s$ in our CPT MDP into several states encoding the state, partial sum of rewards and timestep all at once, i.e. $\tilde{s} = (s, t, \sum_{k=0}^{t-1} r_k)$. We encode all the dynamics of our CPT MDP into our new conventional MDP, and the reward in our new MDP when going from a state $\tilde{s}_a = (s_a, \sigma_a, t_a)$ to another state $\tilde{s}_b = (s_b, \sigma_b, t_a + 1)$ is simply defined as $\mathcal{U}(\sigma_b) - \mathcal{U}(\sigma_a)$. We see that any non-Markovian policy in the CPT MDP can be rewritten as a non-Markovian policy in the new classical MDP and reciprocally, any non-Markovian policy in the new, classical MDP can be rewritten as a non-Markovian policy in the original CPT MDP. Moreover, choosing 1 as a discounting factor in the new MDP, we note that by telescoping sum, the total reward in the new MDP corresponds exactly to the total utility of the sum of rewards in the CPT MDP. Since it is a classical MDP, our new MDP admits a Markovian optimal policy (Puterman, 2014). This optimal policy is at least as good as any other non-Markovian policy in the new MDP with regards to the expected total reward and therefore at least as good as any other non-Markovian policy in the old CPT MDP with regards to *expected utility* of the total reward. When translating this

optimal policy to a policy in the CPT MDP, we notice it only depends on $(s, \sum_{k=0}^{t-1} r_k, t)$, meaning it is indeed an element of $\Pi_{\Sigma,NS}$. This concludes the proof.

### F.4 PROOF OF THEOREM 4

We will prove the following extended version of Theorem 4.

> **Theorem 12.** *Let $\mathcal{U}$ be continuous and (strictly) increasing. The following statements are equivalent:*
>
> 1. *For any MDP, there exists an optimal policy for* (EUT-PO) *in $\Pi_{M,NS}$.*
> 2. *There exists a function $\varphi : \mathbb{R}^2 \to \mathbb{R}$ such that:*
>
> $$\forall x, a, b \in \mathbb{R}, b \neq 0, \mathcal{U}(x+a) - \mathcal{U}(x) = \varphi(a,b)(\mathcal{U}(x+b) - \mathcal{U}(x)).$$
>
> 3. *There exists $\alpha \in \mathbb{R}$ s.t. $\mathcal{U}''(x) = \alpha \mathcal{U}'(x)$ for all $x \in \mathbb{R}$.*
> 4. *There exist $A, B, C \in \mathbb{R}$ s.t. $\mathcal{U}(x) = Ax + B$ or $\mathcal{U}(x) = A + B\exp(Cx)$ for all $x \in \mathbb{R}$.*
> 5. *There exists a function $\mu : \mathbb{R}^2 \to \mathbb{R}$ such that:*
>
> $$\forall y, c, d \in \mathbb{R}, \mathcal{U}(y+c) - \mathcal{U}(c) = \mu(c,d)(\mathcal{U}(y+d) - \mathcal{U}(d)).$$

We prove a series of implications and equivalences. It can be easily verified from combining all these results that $1 \implies 2 \implies 3 \implies 4 \implies 5 \implies 1$, which proves all the equivalences of the theorem. We proceed to prove each one of our implications in the rest of this section.

**Proof of $3 \Leftrightarrow 4$, $4 \Rightarrow 2$, and $4 \Rightarrow 5$.**

The equivalence $3 \Leftrightarrow 4$ is obtained simply by solving the differential equation for one implication and a simple calculation for the other implication. The implications $4 \Rightarrow 2$ and $4 \Rightarrow 5$ follow from simple algebraic verification.

**Proof of $5 \Rightarrow 2$.**

We suppose 5 holds. For any given $a, b \in \mathbb{R}$ such that $b \neq 0$, we define $\varphi(a,b) := \frac{\mathcal{U}(1+a)-\mathcal{U}(1)}{\mathcal{U}(1+b)-\mathcal{U}(1)}$. Notice that this quantity is well defined since $\mathcal{U}$ being (strictly) increasing (and $b \neq 0$) implies that $\mathcal{U}(1+b) - \mathcal{U}(1) \neq 0$. Then, we use 5 to obtain that for every $x \in \mathbb{R}$,

$$\mathcal{U}(x+b) - \mathcal{U}(x) = \mu(x,1)(\mathcal{U}(1+b) - \mathcal{U}(1)). \tag{11}$$

We conclude the proof of the implication by writing:

$$
\begin{aligned}
\mathcal{U}(x+a) - \mathcal{U}(x) &= \mu(x,1)(\mathcal{U}(1+a) - \mathcal{U}(1)) && \text{(again by 5)}\\
&= \varphi(a,b)\mu(x,1)(\mathcal{U}(1+b) - \mathcal{U}(1)) && \text{(using the above definition of } \varphi(a,b))\\
&= \varphi(a,b)(\mathcal{U}(x+b) - \mathcal{U}(x)). && \text{(using Eq. (11))}
\end{aligned}
$$

This shows that 2 holds and concludes the proof.

**Proof of $2 \Rightarrow 4$.** Consider a fixed integer $k$. Let $C_k := \varphi(2 \cdot 2^{-k}, 2^{-k})$ and $u_n := \mathcal{U}(n2^{-k})$ for every $n \in \mathbb{N}$. We have the following recurrence relation for all $n \in \mathbb{N}$:

$$u_{n+2} - u_n = C_k(u_{n+1} - u_n).$$

That is to say for every $n \in \mathbb{N}$,

$$u_{n+2} - C_k u_{n+1} + (C_k - 1)u_n = 0.$$

This recurrence relation can be solved by examining the characteristic polynomial:

$$x^2 - C_k x + (C_k - 1) = 0.$$

The roots are obtained using the quadratic formula: $r_\pm = \frac{C_k \pm \sqrt{C_k^2 - 4(C_k-1)}}{2} = \frac{C_k \pm (C_k - 2)}{2}$.

- If $C_k = 2$, there is only one root, 1, and $\exists D_k, E_k \in \mathbb{R}, \forall n, u_n = (D_k n + E_k)1^n = D_k n + E_k$
- Otherwise, there are two real roots, $(C_k - 1)$ and 1, and $u_n$ is of the form, $u_n = D_k(C_k - 1)^n + E_k \cdot 1^n = D_k C_k^n + E_k$.

This proves that for all $k$, there exists a function in $\{x \mapsto Ax + b, (A, B) \in \mathbb{R}^2\} \bigcup \{x \mapsto A + B \exp(Cx), (A, B) \in \mathbb{R}^2\}$ that coincides with $\mathcal{U}$ on the set $\{\frac{x}{2^k}, x \in \mathbb{N}\}$.

Importantly, all these functions have to be the same (across different values of $k$, i.e. all $D_k$ constants are the same and all $E_k$ constants do also coincide), due to the structure of $\{x \mapsto Ax + b, (A, B) \in \mathbb{R}^2\} \bigcup \{x \mapsto A + B \exp(Cx), (A, B) \in \mathbb{R}^2\}$ and because they all coincide on all the integers with the corresponding value of the same (fixed) utility function $\mathcal{U}$ at the relevant integer. This means that there exists a single function $f$ in $\{x \mapsto Ax + b, (A, B) \in \mathbb{R}^2\} \bigcup \{x \mapsto A + B \exp(Cx), (A, B) \in \mathbb{R}^2\}$ which coincides with $\mathcal{U}$ on all of $\{\frac{x}{2^y}, x \in \mathbb{N}, y \in \mathbb{N}^+\}$. By continuity of $\mathcal{U}$, we obtain that 4 holds.

**Proof of** $4 \Rightarrow 1$**.**

If $\mathcal{U}$ is an affine function $x \mapsto Ax + B$ (for some $A, B \in \mathbb{R}$), then solving the (EUT-PO) problem boils down to solving a traditional MDP in which an optimal Markovian policy always exists (Puterman, 2014).

Let us now assume that the utility function is of the form $x \mapsto A + B \exp(Cx)$ for some $A, B, C \in \mathbb{R}$. Without loss of generality, we can simply ignore the constant $A$ in the optimization problem (EUT-PO) and just assume we are maximizing $\mathcal{U}(x) = B \exp(Cx)$. Recall that we are considering a finite-horizon setting with horizon length $H$. For any $0 \leq T \leq H$, we say that a policy $\pi \in \Pi_{\Sigma, NS}$ is **Markovian in the last** $T$ **steps** if there exists a function $f$ defined from $\mathcal{S} \times \mathbb{N}$ (into the set of policies) such that:

$$\forall \sigma \in \mathbb{R}, \forall t \geq H - T, \forall s \in \mathcal{S}, \pi(s, \sigma, t) = f(s, t).$$

Using again the unwinded MDP construction like in the proof of Proposition 3, we can find a policy $\pi^\star \in \Pi_{\Sigma, NS}^D$ which is "totally" optimal: that is to say, starting from any $(s, \sigma, t)$, $\mathbb{E}\left[\mathcal{U}(\sigma + \sum_{k=t}^{H-1} r_k)\right]$ is maximal when following policy $\pi^\star$. We proceed by induction to prove the assertion $\mathcal{P}_T$: 'There exists a deterministic totally optimal policy $\pi_T$ which is Markovian in the last $T$ steps' for any $T \leq H$, especially for $T = H$ which is the desired result.

**Initialization:** $\mathcal{P}_0$ is true with $\pi_0 = \pi^\star$.

**Induction:** Let us suppose $\mathcal{P}_T$ is true for some $T < H$. We define $\pi_{T+1}$ by:

$$\pi_{T+1}(s, \sigma, t) := \begin{cases} \pi_T(s, 0, t) & \text{if } t = H - T - 1 \\ \pi_T(s, \sigma, t) & \text{otherwise} . \end{cases}$$

We see that $\pi_{T+1}$ is a deterministic policy that is Markovian in the last $T + 1$ steps. We also see that for any $t \geq H - T$ and any $\sigma \in \mathbb{R}, s \in \mathcal{S}$, starting from $(s, \sigma, t)$, $\mathbb{E}\left[\mathcal{U}(\sigma + \sum_{k=t}^{H-1} r_k)\right]$ is maximal when following policy $\pi_{T+1}$. We need to prove it for others values of $t$. i.e. $t \leq H - T - 1$.

Because $\pi_{T+1}$ is Markovian in the last $T + 1$ steps, the probability distribution on future states, actions and rewards starting from $(s, \sigma, H - T - 1)$ does not depend on $\sigma$.

We know that it optimizes $\mathbb{E}\left[\mathcal{U}(0 + \sum_{k=t}^{H-1} r_k)\right]$, and we want to show that it optimizes $\mathbb{E}\left[\mathcal{U}(\sigma + \sum_{k=H-T-1}^{H-1} r_k)\right]$ for all $\sigma \in \mathbb{R}$. This is where we use the form of the utility function to remark that

$$\mathbb{E}\left[\mathcal{U}\left(\sigma + \sum_{k=H-T-1}^{H-1} r_k\right)\right] = \mathbb{E}\left[\exp(C\sigma)\mathcal{U}\left(\sum_{k=H-T-1}^{H-1} r_k\right)\right] = \exp(C\sigma)\mathbb{E}\left[\mathcal{U}\left(\sum_{k=H-T-1}^{H-1} r_k\right)\right]$$

and a maximizer for $\sigma = 0$ is therefore a maximizer for all $\sigma$.

We know now that $\pi_{T+1}$ is optimal starting from any $(s, \sigma, t)$ if $t \geq H - T - 1$. Note that now, step $H - T - 1$ is included.

Starting from $(s, \sigma, t)$ with $t < H - T - 1$, we know that $\pi_T$ maximizes $\mathbb{E}\left[\mathcal{U}(\sigma + \sum_{k=t}^{H-1} r_k)\right]$. We notice, using the tower rule:

$$\mathbb{E}\left[\mathcal{U}\left(\sigma + \sum_{k=t}^{H-1} r_k\right)\right] = \mathbb{E}\left[\mathcal{U}\left(\sigma + \sum_{k=t}^{H-T-2} r_k + \sum_{k=H-T-1}^{H-1} r_k\right)\right]$$

$$= \mathbb{E}\left[\mathbb{E}\left[\mathcal{U}\left(\underbrace{\sigma + \sum_{k=t}^{H-T-2} r_k}_{\sigma'} + \sum_{k=H-T-1}^{H-1} r_k\right) \Big| \underbrace{\sigma + \sum_{k=t}^{H-T-2} r_k}_{\sigma'}, s_{H-T-1}\right]\right] .$$

Because $\pi_{T+1}$ is as good as $\pi_T$ at maximizing $\mathbb{E}\left[\mathcal{U}\left(\sigma' + \sum_{k=H-T-1}^{H-1} \gamma^k r_k\right)\right]$ starting from $(\sigma', S_{H-T-1}, T-H-1)$, we conclude that $\pi_{T+1}$ performs as well as $\pi_T$, because the inner conditional expectation is the same and the first steps are the same.

**Therefore, $\mathcal{P}_{T+1}$ is true.**

**Conclusion:** $\mathcal{P}_H$ is true, which is our desired result.

**Proof of $1 \Rightarrow 2$.**

Let us show $\neg 2 \Rightarrow \neg 1$. $\neg 2$ means that for any function $\varphi : \mathbb{R}^2 \mapsto \mathbb{R}$, there exists $x, a, b \in \mathbb{R}$ such that $b \neq 0$ and $\mathcal{U}(x+a) - \mathcal{U}(x) \neq \varphi(a, b)(\mathcal{U}(x+b) - \mathcal{U}(x))$.

Define now $\varphi : \mathbb{R}^2 \mapsto \mathbb{R}$ by $\varphi(\alpha, \beta) = \frac{\mathcal{U}(\alpha) - \mathcal{U}(0)}{\mathcal{U}(\beta) - \mathcal{U}(0)}$ for all $\alpha \in \mathbb{R}, \beta \neq 0$ and $\varphi(\alpha, \beta) = 1$ for $\beta = 0$. It follows that there exist $x, a \in \mathbb{R}, b \neq 0$ (given by $\neg 2$ above) such that $\mathcal{U}(x+a) - \mathcal{U}(x) \neq \varphi(a, b)(\mathcal{U}(x+b) - \mathcal{U}(x))$.

As a consequence, we obtain $\frac{\mathcal{U}(x+a) - \mathcal{U}(x)}{\mathcal{U}(x+b) - \mathcal{U}(x)} \neq \frac{\mathcal{U}(a) - \mathcal{U}(0)}{\mathcal{U}(b) - \mathcal{U}(0)}$. Our idea now is to exploit this difference in utility to build a situation in which a non-Markovian strategy is clearly more profitable in view of our (EUT-PO) policy optimization problem.

Suppose without loss of generality that $b > a > 0$ and $\frac{\mathcal{U}(x+a) - \mathcal{U}(x)}{\mathcal{U}(x+b) - \mathcal{U}(x)} > \frac{\mathcal{U}(a) - \mathcal{U}(0)}{\mathcal{U}(b) - \mathcal{U}(0)}$ without loss of generality. In the other cases, the inequalities might get reversed but the gist of the proof stays the same. We define $p$ as the halfpoint

$$p := \frac{1}{2}\left(\frac{\mathcal{U}(x+a) - \mathcal{U}(x)}{\mathcal{U}(x+b) - \mathcal{U}(x)} + \frac{\mathcal{U}(a) - \mathcal{U}(0)}{\mathcal{U}(b) - \mathcal{U}(0)}\right) . \tag{12}$$

Since $\mathcal{U}$ is strictly increasing, we have that $p \in (0, 1)$.

We now consider an MDP with three states $s_0, s_1, s_2$ where $s_2$ is a terminal state that leads to nowhere and $s_0$ is the starting state. Whatever action is taken in $s_0$, we transition to $s_1$, with reward $x$ with probability $\frac{1}{2}$ and reward $y$ with probability $\frac{1}{2}$. Once in state $s_1$, we can take action A, which yields reward $a$ with probability 1, or take action $B$, which yields reward $b$ with probaility $p$ and 0 otherwise. Both lead to $s_2$ and the end of the episode with certainty. Here, to maximize the (EUT-PO) objective, one has to adopt a non-Markovian strategy in $s_1$, hence disproving assertion 1. Indeed, knowing the reward achieved in the past step (between states $s_0$ and $s_1$) allows to decide whether to take more risks or not to achieve a higher EUT return.

We elaborate on this claim in what follows. Observe first that

$$\frac{\mathcal{U}(x+a) - \mathcal{U}(x)}{\mathcal{U}(x+b) - \mathcal{U}(x)} > p > \frac{\mathcal{U}(a) - \mathcal{U}(0)}{\mathcal{U}(b) - \mathcal{U}(0)} . \tag{13}$$

What is the best action to choose if we are in state $s_1$ and have had return 0 so far? We know there is a deterministic best action to take. Action 1 yields total reward $\mathcal{U}(a)$. Choosing action 2 yields:

$$\mathbb{E}(\mathcal{U}(r_0 + r_1)|r_0 = 0) = \mathbb{E}(\mathcal{U}(r_1)) = p\mathcal{U}(b) + (1-p)\mathcal{U}(0) = p(\mathcal{U}(b) - \mathcal{U}(0)) + \mathcal{U}(0) > \mathcal{U}(a), \tag{14}$$

where the strict inequality follows from using (13). So it is *strictly* better to choose action 2 over action 1 if we are in $s_1$ and have had a return 0 so far.

What is the best action to choose if we are in state $s_1$ and have had return $x$ so far? We know again that there is a deterministic best action to take. Action 1 yields total reward $\mathcal{U}(x + a)$. Choosing action 2 yields:

$$\mathbb{E}(\mathcal{U}(r_0 + r_1)|r_0 = x) = \mathbb{E}(\mathcal{U}(r_1 + x)) = p\mathcal{U}(b + x) + (1 - p)\mathcal{U}(x)$$
$$= p(\mathcal{U}(b + x) - \mathcal{U}(x)) + \mathcal{U}(x) < \mathcal{U}(a + x), \quad (15)$$

where the strict inequality follows from using again (13). So it is *strictly* better to choose action 1 over action 2 if we are in $s_1$ and have had a return $x$ so far.

We conclude from both cases that there is no optimal Markovian policy.

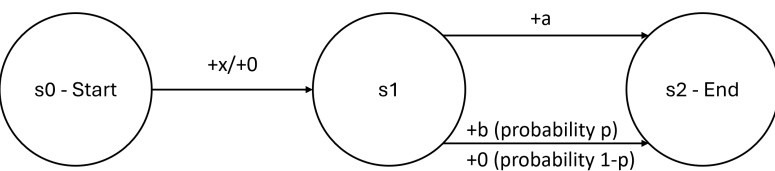

Figure 9: MDP serving as counterexample for the proof of the last implication. While this example has random rewards, another counterexample with random transitions and deterministic rewards can be designed, in the same way as in Remark 11.

### F.5    PROOF OF PROPOSITION 5

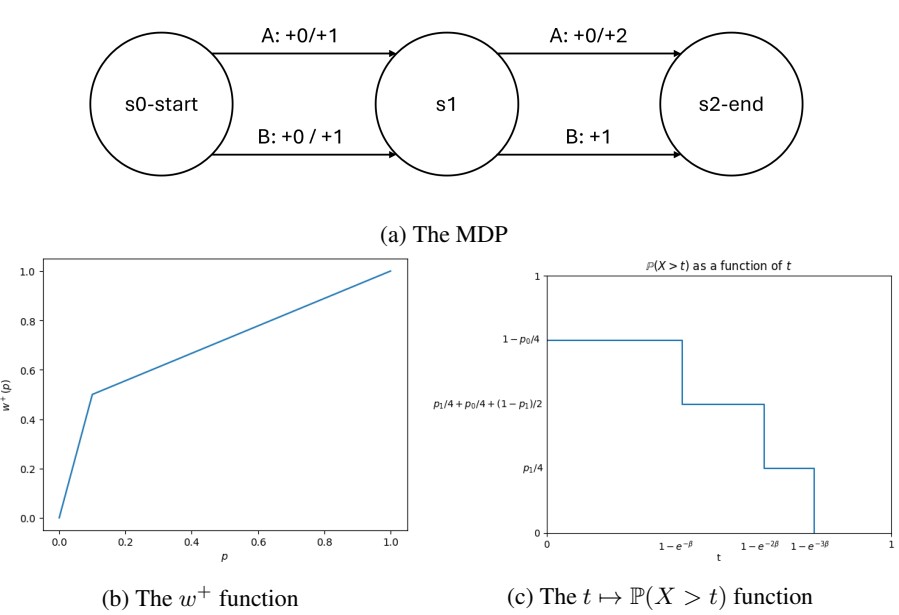

(a) The MDP

(b) The $w^+$ function

(c) The $t \mapsto \mathbb{P}(X > t)$ function

Figure 10: Figures for the proof of Proposition 5

We proceed in the same way as for Proposition 2 by providing a counterexample. We consider the utility function $\mathcal{U} : x \mapsto 1 - \exp(-\beta x)$ with $\beta = \frac{1}{2}$, and the same $w^+$ function as in the proof of Proposition 2:

$$w^+(x) = \begin{cases} 5x & \text{if } x \leq 0.1, \\ \frac{1}{2} + \frac{5}{9}(x - 0.1) & \text{otherwise.} \end{cases}$$

We also set $w^- = 0$. However, we consider another MDP. Our MDP has three states: an initial state $s_0$, an intermediate state $s_1$, and a terminal state $s_2$. There are two actions: A and B. All

trajectories start in $s_0$. Any action from $s_0$ leads to $s_1$ with probability 1 and yields reward $+1$ with probability $\frac{1}{2}$ and 0 otherwise. The action taken when in $s_0$ is completely irrelevant. Any action taken in $s_1$ leads to $s_2$ with probability 1 and the episode stops as soon as $s_2$ is reached. When taking action A in $s_1$, the reward is either 0 or $+2$, with probability $\frac{1}{2}$ each. When taking action B in $s_1$, the reward is $+1$ with probability 1. All policies in $\Pi_{NM}$ can be described by $(p_{\text{start}}, p_0, p_1)$, where $p_{\text{start}}$ is the probability of choosing action A when in $s_0$, $p_0$ is the probability of choosing action A in $s_1$ if the transition from $s_0$ to $s_1$ yielded reward 0 and $p_1$ is the probability of choosing action A in $s_1$ if the transition from $s_0$ to $s_1$ yielded reward 1. $p_{\text{start}}$ is irrelevant to the performance of the policy so we can ignore it. The set of Markovian policies here is the set of policies such as $p_0 = p_1$. $\mathbb{C}(\pi)$ is a piecewise affine function of $p_0$ and $p_1$ and it can therefore be directly maximized. We omit the calculations here: one can check that the best achievable CPT value for Markovian policies is $\approx 0.616$ for $p_0 = p_1 = 0.4$ but that a CPT value of $\approx 0.625$ is achievable for $p_0 = 0$ and $p_1 = 0.4$, proving the lemma.

# G    PROOFS AND ADDITIONAL DETAILS FOR SECTION 4

## G.1    PROOF OF THEOREM 6

The CPT value is a difference between two integrals (see definition in (1)). In what follows, we compute the derivative of the first integral assuming that the second one is zero in the CPT value. A similar treatment can be applied to the second integral. We skip these redundant details for conciseness.

**Remark 13.** *As we consider a finite horizon setting with finite state and action spaces, the integral on trajectories $\tau$ are in fact finite sums, allowing us to differentiate freely. We still write the proof with $\int$ signs, signalling our hope that, under some technical assumptions, our proof could be generalized to a setting with infinite horizon and/or infinite state and action spaces.*

Using the shorthand notation $X = \sum_{t=0}^{H-1} r_t$, we first observe that:

$$\mathbb{C}(X) = \int_{z=0}^{+\infty} w(\mathbb{P}(\mathcal{U}(X) > z)dz = \int_{z=0}^{+\infty} w\left(\int_{\tau \text{ such as } \mathcal{U}(R(\tau))>z} \rho_\theta(\tau)d\tau\right)dz\,, \quad (16)$$

where $\rho_\theta$ is the trajectory probability distribution induced by the policy $\pi_\theta$ defined for any $H$-length trajectory $\tau = (s_0, a_0, \cdots, s_{H-1}, a_{H-1})$ as follows:

$$\rho_\theta(\tau) = p(s_0) \prod_{t=0}^{H-1} \pi_\theta(a_t|h_t)p(s_{t+1}|h_t, a_t)\,. \quad (17)$$

**Remark 14.** *Recall that we have ignored the second integral in the CPT value definition for conciseness.*

Starting from the above expression (16), it follows from using the chain rule that:

$$\nabla_\theta \mathbb{C}(X) = \int_{z=0}^{+\infty} w'(\mathbb{P}(\mathcal{U}(X > z))\nabla_\theta\left(\int_{\tau \text{ such as } \mathcal{U}(R(\tau))>z} \rho_\theta(\tau)d\tau\right)dz \quad (18)$$

$$= \int_{z=0}^{+\infty} w'(\mathbb{P}(\mathcal{U}(X > z)) \int_{\tau \text{ such as } \mathcal{U}(R(\tau))>z} \nabla_\theta\rho_\theta(\tau)d\tau dz \quad (19)$$

$$= \int_\tau \int_{z=0}^{\mathcal{U}(R(\tau))} w'(\mathbb{P}(\mathcal{U}(X) > z))\nabla_\theta\rho_\theta(\tau)dz d\tau \quad (20)$$

$$= \int_\tau \phi(\mathcal{U}(R(\tau)))\nabla_\theta\rho_\theta(\tau)d\tau\,, \quad (21)$$

where $\phi(t) := \int_{z=0}^{t} w'(\mathbb{P}(\mathcal{U}(X) > z))dz$ for any real $t$.

We now use the standard log trick to rewrite our integral as an expectation:

$$\nabla_\theta \mathbb{C}(X) = \int_\tau \phi(\mathcal{U}(R(\tau)))\rho(\tau)\nabla_\theta \log \rho(\tau)d\tau = \mathbb{E}_{\tau\sim\rho}[\phi(\mathcal{U}(R(\tau)))\nabla_\theta \log \rho(\tau)]\,.$$

Furthermore, we can expand the gradient of the score function using (17) as follows:

$$\log \rho_\theta(\tau) = \log p(s_0) + \sum_{t=0}^{H-1} \log \pi_\theta(a_t|h_t) + \sum_{t=0}^{H-1} \log p(s_{t+1}|h_t, a_t), \tag{22}$$

$$\nabla_\theta \log \rho_\theta(\tau) = \sum_{t=0}^{H-1} \nabla_\theta \log \pi_\theta(a_t|h_t), \tag{23}$$

where the last step follows from observing that only the policy terms involve a dependence on the parameter $\theta$. Combining (21) and (23) leads to our final policy gradient expression:

$$\nabla_\theta \mathbb{C}(X) = \mathbb{E}\left[\phi\left(\sum_{t=0}^{H-1} r_t\right) \sum_{t=0}^{H-1} \nabla_\theta \log \pi_\theta(a_t|h_t)\right]. \tag{24}$$

Note that we have used the notation $\phi$ above instead of $\varphi$ used in Theorem 6 to avoid the confusion with the full definition of $\varphi$ which involves both integrals.

## G.2 ALTERNATIVE PRACTICAL PROCEDURE FOR COMPUTING STOCHASTIC POLICY GRADIENTS

In this section, we discuss an alternative approximation procedure to the one proposed in section 4 for computing stochastic policy gradients. More precisely, we seek to approximate $\varphi(R(\tau))$ without the need for estimating quantiles and using order statistics for this. This alternatively procedure will be especially useful in practice when the probability distortion $w$ is not necessarily differentiable or smooth. As discussed in the main part, one of the key challenges to compute stochastic policy gradients is to compute the integral terms appearing in the policy gradient expression. Our idea here is to approximate the probability distortion function $w$ by a piecewise (linear or quadratic) function, leveraging the following useful lemma which shows that the integral is simple to compute when $w$ is quadratic for instance.

**Lemma 15.** *Let $X$ be a real-valued random variable and suppose that the weight function $w$ is quadratic on an interval $[a, b]$ for some positive constants $a, b$, hence there exist $\alpha, \beta \in \mathbb{R}$ s.t. for all $x \in [a, b], w'(x) = \alpha x + \beta$. Let $Y_{a,b} := \min(\max(\mathcal{U}(X) - a, b - a), 0)$. Then, we have that $\int_a^b w'(\mathbb{P}(\mathcal{U}(X) > z))dz = \alpha \mathbb{E}[Y_{a,b}] + \beta(b - a)$.*

*Proof.* For any $a, b \in \mathbb{R}$ s.t. $a \leq b$, we have

$$\int_a^b w'(\mathbb{P}(\mathcal{U}(X) > z))dz = \int_0^{b-a} w'(\mathbb{P}(\mathcal{U}(X) - a > v))dv$$

$$= \int_0^{b-a} (\alpha(\mathbb{P}(\mathcal{U}(X) - a > v)) + \beta)dv$$

$$= \alpha \int_0^{b-a} \mathbb{P}(\mathcal{U}(X) - a > v)dv + \beta(b - a)$$

$$= \alpha \int_0^{b-a} \mathbb{P}(Y_{a,b} > v)dv + \beta(b - a)$$

$$= \alpha \int_0^{+\infty} \mathbb{P}(Y_{a,b} > v)dv + \beta(b - a)$$

$$= \alpha \mathbb{E}[Y_{a,b}] + \beta(b - a).$$

$\square$

This result is convenient: Instead of estimating an entire probability distribution, we just have to estimate an expectation, which is much easier. However, we cannot reasonably approximate an arbitrary weight function by a quadratic function. Therefore, we consider the larger class of piecewise quadratic functions for which Lemma 15 extends naturally.

**Proposition 16.** *Let $w$ be piecewise quadratic: there exists $q_1 < q_2 < .... < q_k$, with $q_1 = 0$ and $q_k = 1$, as well as reals $\alpha_1, ...., \alpha_k$, $\beta_1, ...., \beta_k$ and $\delta_1, ...., \delta_k$ such as $w(x) = \sum_{i=1}^{k-1} \mathbb{1}_{[q_i, q_{i+1}[}(t)(\frac{1}{2}\alpha_i t^2 + \beta_i t + \delta_i)$. For all $1 \leq i \leq k-1$, define the $i$-th quantile of $\mathcal{U}(X)$ as $\tilde{q}_i := \sup\{t \in \mathbb{R} \cup \{+\infty, -\infty\}, \mathbb{P}(\mathcal{U}(X) > t) \geq q_i\}$. Then, for any given $t \in [\tilde{q}_{j+1}, \tilde{q}_j[$:*

$$\int_0^t w'(\mathbb{P}(U(X) > z))dz = \sum_{i=j+1}^{k-1} \left(\alpha_i \mathbb{E}(Y_{\tilde{q}_{i+1}, \tilde{q}_i}) + \beta_i(\tilde{q}_i - \tilde{q}_{i+1})\right) + \alpha_j \mathbb{E}(Y_{\tilde{q}_j, t}) + \beta_j(t - \tilde{q}_{j+1}).$$

*Proof.* We simply apply Lemma 15 to each segment:

$$\int_0^t w'(\mathbb{P}(U(X) > z))dz = \sum_{i=j+1}^{k-1} \int_{\tilde{q}_{i+1}}^{\tilde{q}_i} w'(\mathbb{P}(U(X) > z))dz + \int_{\tilde{q}_{j+1}}^t w'(\mathbb{P}(U(X) > z))dz$$

$$= \sum_{i=j+1}^{k-1} \left(\alpha_i \mathbb{E}(Y_{\tilde{q}_{i+1}, \tilde{q}_i}) + \beta_i(\tilde{q}_i - \tilde{q}_{i+1})\right) + \alpha_j \mathbb{E}(Y_{\tilde{q}_j, t}) + \beta_j(t - \tilde{q}_{j+1}).$$

$\square$

The above lemma shows that we would have to estimate several quantiles and expectations to use this result. In particular, the expectation $\mathbb{E}(Y_{\tilde{q}_j, t})$ introduces some undesired computational complexity as the term differs for every $t$. However, if we rather consider a simpler piecewise affine approximation of $w$ which can be computed once before any computation (independently from the rest) if the probability distortion function $w$ is priorly known (which we implicitly suppose throughout this work), the expression is greatly simplified, yielding Lemma 17.

**Lemma 17.** *Suppose that the weight function $w : [0,1] \mapsto [0,1]$ is piecewise affine, i.e. there exists $q_1 < q_2 < .... < q_k$, with $q_1 = 0$ and $q_k = 1$, as well as reals $\beta_1, ...., \beta_k$ and $\delta_1, ...., \delta_k$ s.t. $w(x) = \sum_{i=1}^{k-1} \mathbb{1}_{[q_i, q_{i+1}[}(x)(\beta_i x + \delta_i)$ for any $x \in [0,1]$. Let $\tilde{q}_i := \sup\{t \in \mathbb{R} \cup \{+\infty, -\infty\}, \mathbb{P}(\mathcal{U}(X) > t) \geq q_i\}$ for any $i = 1, \cdots, k$. Then for any $1 \leq j \leq k-1$ and any $t \in [\tilde{q}_{j+1}, \tilde{q}_j[$,*

$$\int_0^t w'(\mathbb{P}(\mathcal{U}(R(\tau)) > z))dz = \sum_{i=j+1}^{k-1} \left(\beta_i(\tilde{q}_i - \tilde{q}_{i+1})\right) + \beta_j(t - \tilde{q}_{j+1}).$$

# H   MORE DETAILS ABOUT SECTION 5 AND ADDITIONAL EXPERIMENTS

| Environment | Utility function | $w^+$ function | Figure | Comment |
|---|---|---|---|---|
| Grid | Various | 3-segment piecewise affine function | Fig. 2 and Apdx H.4 | We observe convergence and various behaviours for various utility functions |
| Traffic Control | Identity | Risk averse ($w_{ra}$) | Apdx. H.5 | The policy goes around the city center |
| Traffic Control | Identity | Risk neutral (Id.) | Apdx. H.5 | The policy goes through the city center |
| Traffic Control | Identity | Risk averse ($w_{ra}$) / Risk-neutral (Id.) | Apdx. H.5 | Same behavior, entropy regularization needed |
| Scalable Grid | Identity | Risk-averse ($w_{ra}$) | Fig. 3 | Our algorithm converges faster than CPT-SPSA-G for large grids |
| Electricity Management | Identity | Very risk averse ($w_{vra}$) / Very risk seeking ($w_{vrs}$) / Risk-neutral (Id.) | Fig. 2 | Convergence to different reward distributions in accordance to behavior to risk |
| Fig. 14 | Exponential, Kahneman-Tversky | Risk-neutral (Id.) | Fig. 18 | The result illustrates Theorem 4 |
| Fig. 12a | Identity | Risk-seeking ($w_{rs}$) | Fig. 13 | The result illustrates Proposition 2 |

Table 2: Summary of experiments

Table 2 recaps the various experimental settings. The risk-neutral $w^+$ function is simply the identity function. As for the definition of other probability distortion functions $w^+$ we use for experiments, we define:

$$w_{ra}(x) := \begin{cases} 0.5x & \text{if } x \leq 0.9, \\ 5.5x - 4.5 & \text{otherwise.} \end{cases} \quad w_{rs}(x) := \begin{cases} 5x & \text{if } x \leq 0.1, \\ \frac{1}{2} + \frac{5}{9}(x - 0.1) & \text{otherwise.} \end{cases}$$

$$w_{sra}(x) := \begin{cases} 0.1x & \text{if } x \leq 0.9, \\ 9.1x - 8.1 & \text{otherwise.} \end{cases} \quad w_{srs}(x) := \begin{cases} 9x & \text{if } x \leq 0.1, \\ \frac{1}{9}x + \frac{8}{9} & \text{otherwise.} \end{cases}$$

Instead of vanilla stochastic gradient descent, we use the Adam optimizer to speed up convergence. In our Python implementation, we use the same batch of trajectories for estimating the function $\varphi$ and for the performing the stochastic gradient ascent step. We have run the experiments on a laptop with a 13th Gen Intel Core i7-1360P2.20 GHz CPU and 32 GB of RAM.

We use the tanh activation function before the last softmax layer to encourage exploration and reduce the risk of converging to local optima which may occasionally occur for some runs.

## H.1 ADDITIONAL FIGURE

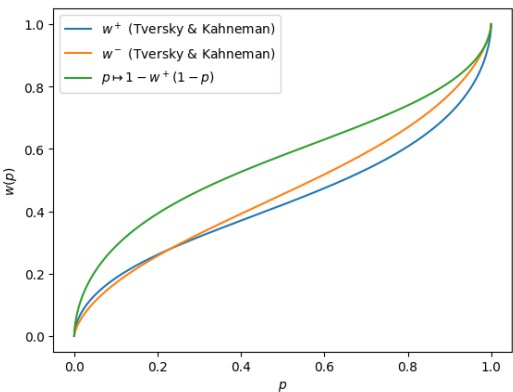

Figure 11: Illustration of the flexibility of CPT compared to the Distortion Risk Measure. Notice how $w^-$ is distinct from both $w^+$ and $p \mapsto 1 - w^+(1-p)$.

## H.2 ILLUSTRATION OF PROPOSITION 2: ABOUT THE NEED FOR STOCHASTIC POLICIES IN CPT-RL

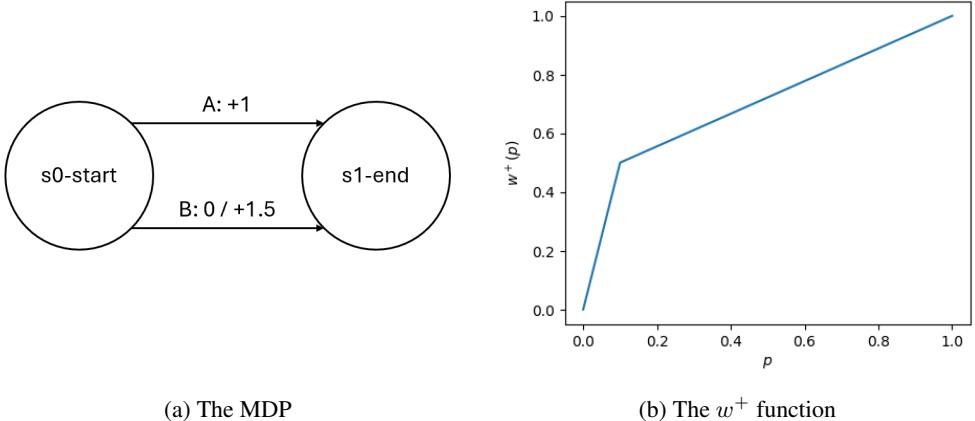

(a) The MDP        (b) The $w^+$ function

Figure 12: Setting of the experiment on non-deterministic policies and batch size influence

We illustrate Proposition 2 and study experimentally the behavior of our algorithm with regards to small batch sizes.

**Setting.** We use the barebones setting (Figure 12) introduced in the proof of Proposition 2 with its $w$ function that aggressively focuses on the 10% of favorable outcomes. Denoting by $p$ the probability of choosing A for a given policy, we look at the value of $p$ at convergence (1000 optimization steps) for various batch sizes. The optimal policy corresponds to $p = 0.8$.

**Insights.** For each batch size we test, we run and plot a hundred training rounds (Figure 13). We fist observe that the policy we obtain with our algorithm indeed approaches the optimal $p = 0.8$ policy.

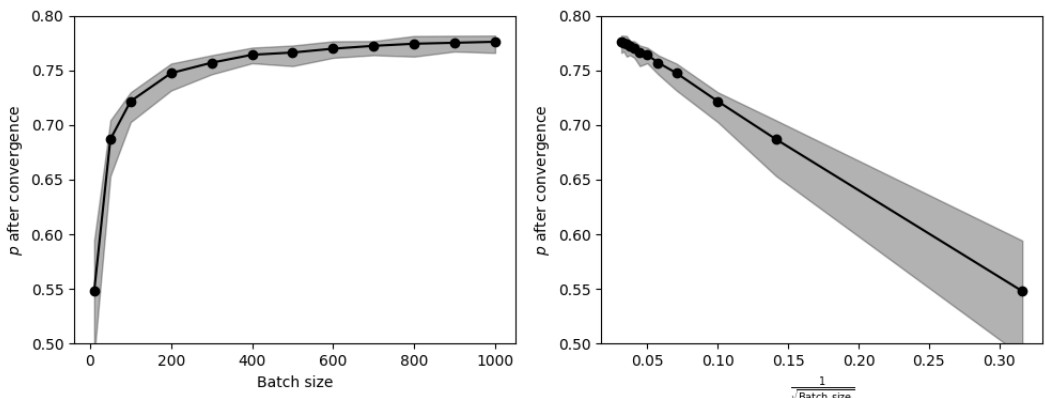

Figure 13: Results of the experiment on non-deterministic policies and batch size influence, over 100 runs. The black dots are the medians and the shaded area represents the interquartile range.

The estimation error (w.r.t. the optimal theoretical value of $p = 0.8$) appears to be of order $\frac{1}{\sqrt{\text{batch size}}}$. It was to be expected that a small batch size would lead to a bias in CPT value and CPT gradient estimation, and, finally, in policy, as a small batch size renders impossible an accurate estimation of the probability distribution of the total return function. The fact that this bias appears to be proportional to the inverse of the square root of the batch size is in line with the standard statistical intuition (as e.g. per the central limit theorem). In our particular example, the estimated $p$ is below (and not above) the theoretical $p$. This is likely because our $w$ function places a strong weight on the top 10% of outcomes. Hence there is an imbalance between the impact of overestimating and underestimating the proportion of good outcomes in a given run: if we underestimate the probability of getting $+1.5$ with a given policy due to sampling, the effect will be stronger than the opposite effect we would get by overestimating the probability of the same error. As the batch size grows, the estimation error is reduced and the effect vanishes.

### H.3 ILLUSTRATION OF THEOREM 4: MARKOVIAN VS NON-MARKOVIAN POLICIES FOR CPT-RL

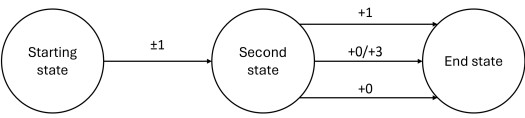

Figure 14: The environment for the experiment on non-Markovian policy

To illustrate the fundamental difference between memoryless utility functions studied in Theorem 4 and the others we conduct a small experiment on a simple setting (Figure 14), similar to the one introduced in the proof of the theorem. We consider three states and three actions. From the starting state, any action leads to the second state with probability 1 and yields a reward of $+1$ with probability $\frac{1}{2}$ and of $-1$ with probability $\frac{1}{2}$. Once in the second state, the first action yields reward $+1$ with probability 1, the second action yields 0 or 3 with probability $\frac{1}{2}$ each, and the third action always yields 0. We compare the performance of a policy parametrized in $\Pi_{\Sigma,NS}$ and one in $\Pi_{M,NS}$.

**Insights.** The results (Figure 18) illustrate indeed the performance advantage of the non-Markovian policy compared to the Markovian one in the case of a non-affine, non-exponential utility function, and the absence thereof in the exponential setting.

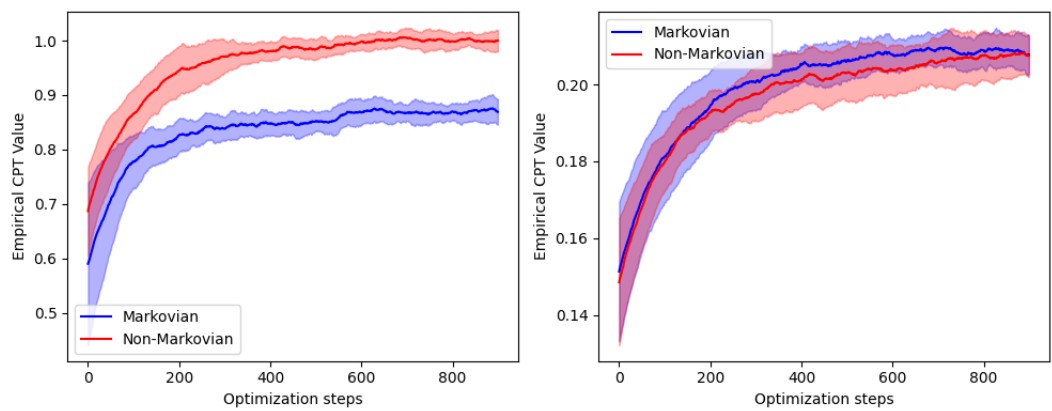

Figure 15: Comparison of Markovian and Non-Markovian policy performances for non-exponential (left) and exponential (right) utility functions. Shaded areas represent a range of $\pm$ one standard deviation over 20 runs.

## H.4 GRID ENVIRONMENT

| $\downarrow$ | $\downarrow$ | $\downarrow$ | $\downarrow$ |
|---|---|---|---|
| $\rightarrow$ | $\downarrow$ | $\downarrow$ | $\downarrow$ |
| $\rightarrow$ | $\rightarrow$ | $\rightarrow$ | $\downarrow$ |
| +5 | $\rightarrow$ | $\rightarrow$ | +6 |

(a) A risk-neutral optimal policy obtained with our algorithm

| $\downarrow$ | $\downarrow$ | $\downarrow$ | $\downarrow$ |
|---|---|---|---|
| $\downarrow$ | $\downarrow$ | $\downarrow$ | $\downarrow$ |
| $\downarrow$ | $\rightarrow$ | $\rightarrow$ | $\downarrow$ |
| +5 | $\rightarrow$ | $\rightarrow$ | +6 |

(b) An optimal policy obtained by training with the risk-averse utility $\mathcal{U} : x \mapsto \sqrt{x}$

Figure 16: Comparison of optimal policies under risk-neutral and risk-averse scenarios

**Exploration.** To avoid our gradient ascent algorithm getting stuck in a local optimum, we have to ensure enough exploration is going on. Therefore, we tweak the last layer of the neural network to prevent every action's probability from vanishing too soon. We choose a parameter $\alpha$, choose our last layer as $x \mapsto \text{softmax}(\alpha \tanh(x/\alpha))$, and we let $\alpha$ slowly grow with the iterations. A small $\alpha$ forces exploration, larger $\alpha$ allows for more exploitation: this is similar to an $\epsilon$-*greedy* scheme (with $\epsilon$ decaying as $\alpha$ grows), as it forces every action to be chosen with at least a small probability.

## H.5 TRAFFIC CONTROL

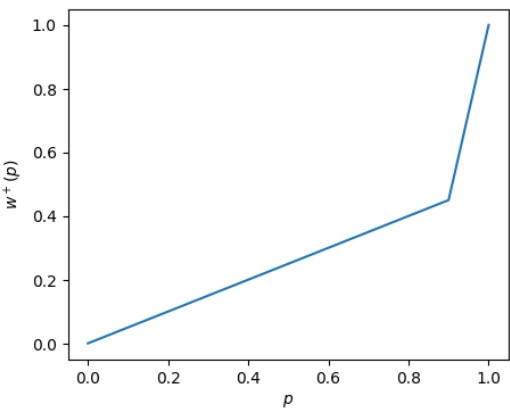

Figure 17: The probability distortion function $w^+$ used for the traffic control experiment.

(a) Training with our $w$ function for traffic control ($3 \times 3$)

(b) Risk-neutral reference

(c) Training with our $w$ function for traffic control ($4 \times 4$)

(d) Risk-neutral reference

Figure 18: Examples of policies obtained with our algorithm. Question marks indicate a non-deterministic action selection in a given state.

**Implementation details.** In both cases, the risk-neutral optimal solution (going around the city center) is also a local optimum for the risk-averse objective, and, because it is a shorter path, is easier to stumble upon by chance when exploring the MDP. This means we have to implement special measures to force exploration. The algorithm used *as is* is prone to get stuck from time to time in local minima on this example. It would seem that our $w$ function, which is aggressively risk-averse, hinders exploration. To mitigate this, we introduce an entropy regularization term that we add to the score function with a decaying regularization weight in the policy gradient found in Theorem 6, see appendix H for further details. We incorporate entropy regularization in the policy gradient as follows:

$$\mathbb{E}\left[ \varphi \left( \sum_{t=0}^{H-1} r_t \right) \sum_{t=0}^{H-1} \nabla_\theta \Big( \log \pi_\theta(a_t|s_t) + \underbrace{\alpha_n H(\pi_\theta(a_t|s_t))}_{\text{Entropy regularization term}} \Big) \right], \tag{25}$$

where $\alpha_n$ is the weight of the regularization. We found that a decaying $\alpha_n$ yielded the best results.

On the $4 \times 4$ grid, we also start by pretraining our model with a risk-neutral method for a few steps, to accelerate training and avoid some bad local optima we can stumble upon due to unlucky policy initialization, before carrying on with our risk-aware method.

## H.6 ELECTRICITY MANAGEMENT

Public data is available online.[4]

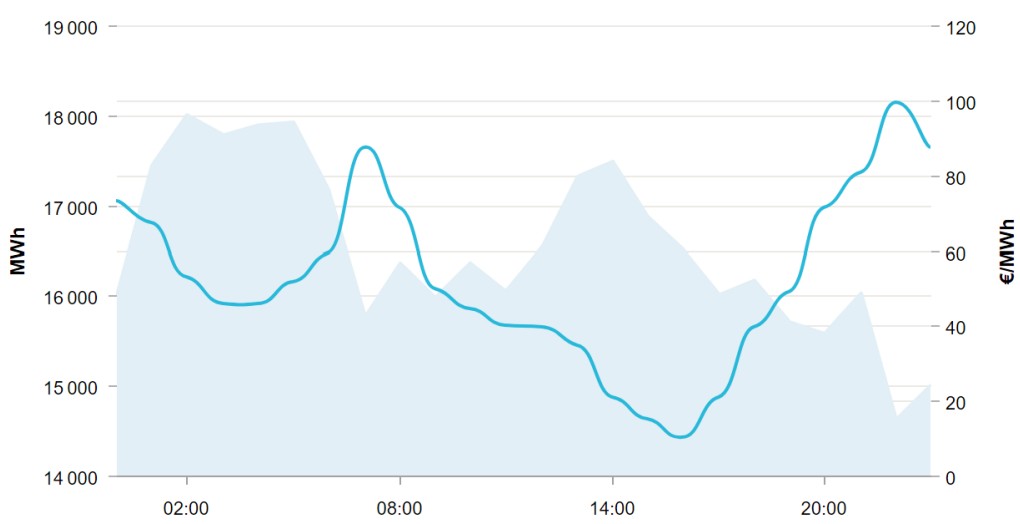

Figure 19: Electricity prices in a typical day, the blue line (right-hand side scale) records the electricity price on the European market, the shaded area (left-hand side scale) represents the total electricity production in France.

---

[4] www.services-rte.com/en/view-data-published-by-rte/
france-spot-electricity-exchange.html

## H.7 Trading in Financial Markets

We discuss here an application of our methodology to financial trading. The goal is to train RL trading agents using our general PG algorithm in the setting of our CPT-RL framework.

**Environment: general description.** We consider a gym trading environment available online, all the details about this environment are available here: `https://gym-trading-env.readthedocs.io/en/latest/`. This environment simulates stocks and allows to train RL trading agents. For the interest of the reader, we provide a brief summary explaining how the environment works. The environment is build from a given dataframe and a list of possible positions. The dataframe contains market data throughout a given period. The list of possible positions will represent the set of possible actions the agent can take, We provide more details about our specific environment in the following paragraph.

**Our trading environment.** We use data from the Bitcoin USD (BTC-USD) market between May 15th 2018 and March 1st 2022 available in the aforementioned website. We note that the data used follows the same pattern as publicly available data after a few preprocessing steps, the reader can find such data examples at `https://finance.yahoo.com/quote/BTC-USD/history` including the date, a few extracted features ('open', 'high', 'low', 'close') which respectively represent the open price, i.e. the price at which the first trade occurred for the asset at the beginning of the time period, the highest, lowest and last such prices, and the volume in USD which is the total value of all trades executed in a given time period. In particular, we will consider static features (computed once at the beginning of the data frame preprocessing) and dynamic features (computed at each time step) such as the last position taken as introduced by the Gym Trading Environment.

- State space: We consider a seven dimensional continuous state space. Features are constructed from the raw stock market data as previously explained. State transitions are described using the provided time series. See the publicly available code of the environment for more details.

- Action space: We consider three classical types of positions the trader can take in a financial market: SHORT, OUT and LONG. These positions constitute the set of actions. These actions refer to whether the trader expects the price of an asset to rise or fall and how they are positioned to profit from that fluctuation. Extending this setting to a setting with a larger set of positions is straightforward as the environment implementation also supports more complex positions.

- Rewards: The rewards we consider are given by the log values of the ratio of the portfolio valuations at times $t$ and $t-1$. Borrowing interest rates and trading fees are also considered in the computation. The reward function can also be easily modified in the environment thanks to the implementation of the Gym Trading Environment which builds on the standard Gym environments.

**Remark 18.** *One can easily build their own environment by downloading their own dataframe for any historical stock market data and performing their desired preprocessing as for the features they would like to consider to build their states.*

**Experimental setting.** We have tested several utility and probability weighting functions including a risk averse exponential of the form $x \mapsto \frac{1}{\beta}(1 - \exp(-\beta x))$ with different values of $\beta$ as well as the KT (Kahneman and Tversky) function as defined in the main part with different values of the reference point $x_0$ to illustrate its influence.

**Hyperparameters.** We used the following set of parameters to conduct the experiments:

Table 3: Hyperparameters

| Hyperparameter | Value |
|---|---|
| Optimizer | Adam |
| Learning rate | 0.05 |
| Number of episodes | 100 |
| Batch size | 5 |
| Number of steps per episode | 25 |

Additional hyperparameters used are directly reported in the legends of the figures below.

**Results.** We refer the reader to Fig. 20 and Fig. 21 below. We make a few observations:

- Influence of the reference point: It can be seen that the reference point shifts the values of the achieved CPT returns: The smaller the reference point, the larger are the returns. This is because only values larger than the reference point are perceived as positive returns given the definition of the KT utility. This illustrates how the subjective perception of the agent of the returns is taken into account by the model.

- Different return trajectories for different risk averse functions: Different values of $\beta$ lead to different trajectories overall which can translate to different levels of risk aversion. In particular, the curves do not match the identity utility case in the first episodes and show more or less risk taken towards optimizing the CPT returns.

- Influence of the parameter $\alpha$ in KT's utility (Fig. 21): Observe that the exponent $\alpha$ in the utility distorts the function and shifts the returns significantly. Lower values of $\alpha$ lead to higher returns in this setting where the returns (as per the ratio definition of the reward) are smaller than 1. This parameter $\alpha$ provides a degree of freedom to model the behavior of the agent as per their perception of the returns. Different values of $\alpha$ modify the curvature of the utility function (w.r.t. the reference point which is $x_0 = 0$ here) which is concave for gains and convex for losses.

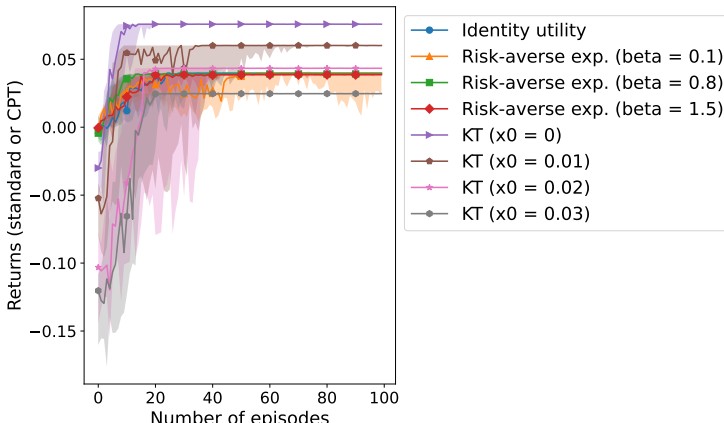

Figure 20: Performance of our PG algorithm on a financial trading application. KT refers to Kahneman and Tversky's utility function, x0 is the reference point used in that utility, exp. refers to exponential. Shaded areas are interquantile (25-75%) margins and curves report the median values over 10 different runs.

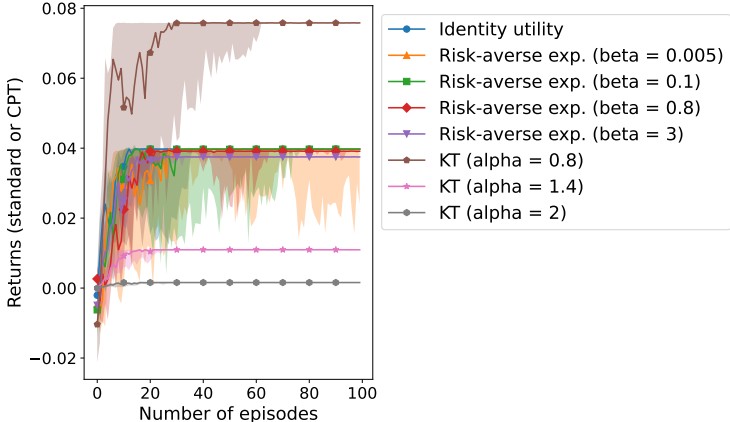

Figure 21: Performance of our PG algorithm on the same financial trading application. KT refers to Kahneman and Tversky's utility function, alpha is the parameter used in the definition of KT's utility, exp. refers to exponential. Shaded areas are interquantile (25-75%) margins and curves report the median values over 10 different runs.

## H.8 Control on MuJoCo Environments

In this section we test our algorithm on the INVERTEDPENDULUM-V5 environment (Todorov et al., 2012) to demonstrate that our PG algorithm is also applicable to other control benchmarks with continuous state and action spaces.

**Hyperparameters.** We used the following set of parameters to obtain our results:

Table 4: Hyperparameters

| Hyperparameter | Value |
|----------------|-------|
| Optimizer | Adam |
| Learning rate | 1e-4 |
| Number of episodes | 2000 |
| Batch size | 32 |
| Maximum number of steps per episode | 200 |

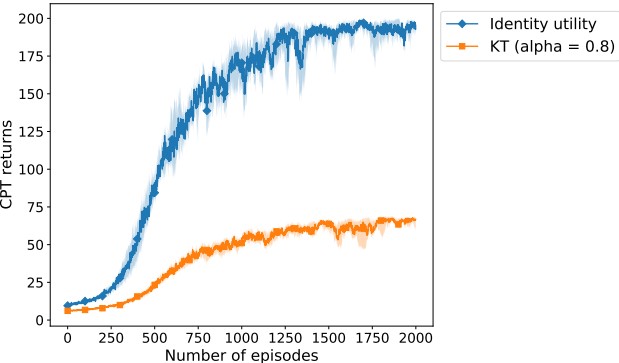

Figure 22: Performance of our PG algorithm on the INVERTEDPENDULUM-V5 environment (Todorov et al., 2012). KT refers to Kahneman and Tversky's utility function, alpha is the parameter used in the definition of KT's utility, exp. refers to exponential. Shaded areas are interquantile (25-75%) margins and curves report the median values over 10 different runs. All the CPT return curves are obtained with the same probability weighting function $w$ which is piecewise affine with three segments (hence different from the standard RL identity setting).

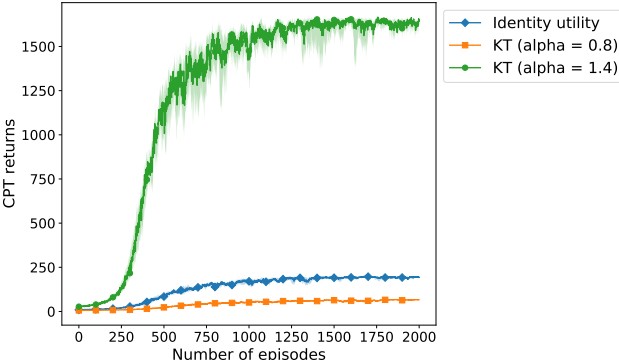

Figure 23: Performance of our PG algorithm on the INVERTEDPENDULUM-V5 environment (Todorov et al., 2012). This figure complements Fig 22 with the CPT returns using a KT utility with $\alpha = 1.4$. Notice that a much higher CPT return is achieved in that case. We also provide Fig. 22 for scaling purposes, the CPT returns being much higher for KT ($\alpha = 1.4$).

