# OpenReview forum: "Beyond Expected Returns: A Policy Gradient Algorithm for Cumulative Prospect Theoretic Reinforcement Learning"
_ICLR.cc/2025/Conference — Submitted to ICLR 2025_

### Official Review · Reviewer_aLSN · 2024-10-28

**Soundness:** 4
**Presentation:** 3
**Contribution:** 3
**Rating:** 6
**Confidence:** 3

**Summary:**

This paper makes three key contributions to the field of cumulative-prospect-theory (CPT)-based reinforcement learning, a framework where the agent aims to maximize not the raw cumulative reward but a transformed version of it. This transformation involves both a utility function and a probability weighting function, introduced to better reflect human value assessments of rewards (e.g., monetary amounts converted into perceived value). The objective is to find a policy that maximizes this transformed cumulative reward.

CPT-based reinforcement learning generalizes various prior approaches, including: (1) the standard RL framework, (2) risk-sensitive RL with an exponential utility function (e.g., as in Noorani et al., 2022), (3) optimization based on VaR/CVaR of the total reward, and (4) the case studied by Kahneman and Tversky (1992).

The first contribution presents new insights into the characteristics of optimal policies within CPT-based reinforcement learning. These findings include: (1) deterministic, history-dependent optimal policies may not always exist; (2) in the absence of weighting functions, a deterministic, non-stationary optimal policy exists that depends on the current state and past cumulative reward; and (3) without weighting functions, a Markovian, non-stationary optimal policy exists if and only if the utility function is either affine or exponential.

The second contribution introduces a policy gradient theorem and a novel algorithm for CPT-based reinforcement learning.

The third contribution is an empirical evaluation of the proposed algorithm. This study includes (1) a performance comparison with CPT-SPSA-G, an existing RL algorithm for CPT problems, across various grid-world environments, demonstrating the proposed algorithm's advantages, particularly in larger environments; and (2) an exploration of different behaviors under varying utility and weighting functions.

**Strengths:**

The theoretical and algorithmic contributions are original and clear. If one believes that CPT-based RL problems are important, then these contributions are also significant.

The paper did a very nice job relating its studied problem with other related problem settings studied previously, making it easy for readers to understand the position of this paper in the literature.

The paper's presentation is clear.

**Weaknesses:**

While I appreciate the contributions the authors made to CPT-based RL problems, I am not convinced of the importance of CPT-based RL problems by reading this paper. The paper cites another paper "... this is particularly important in applications directly involving humans
in the loop such as e-commerce, crowdsourcing and recommendation to name a few" to argue that this class of problems has important applications. However, it is not clear from the cited paper whether other formulations that take into account human behavior can also handle these applications well. In fact, compared with CPT, I feel that a more principled way that considers human behavior is trajectory-based reward RL problems (one reward for the entire trajectory) with human preferences, like RLHF in large language models. While CPT assumes a structure of the final reward (the final metric to optimize involves a weight function, a utility function, and a summation of rewards), these weight functions are unknown and must be learned from data. In contrast, in trajectory-based reward RL problems, there is no assumption on the structure of the metric being optimized. And the metric is learned with human preference data. I wonder how authors think about the pros and cons of CPT in terms of its applications compared to trajectory-based reward RL problem settings.

Another weakness is related to the empirical study of the paper. The experiments only demonstrate the effectiveness of the proposed algorithm but do not include those justifying the necessity of the CPT problem. It would be much better if the authors could demonstrate that in some applications, the data shows that human behavior can be accurately predicted with CPT but not other theories.

**Questions:**

See weaknesses.

---

> ### Author Response · Authors · 2024-11-21
> **Response to reviewer aLSN**
>
> We thank the reviewer for their time, for their valuable feedback and thoughtful comments. We appreciate that the reviewer is faithfully and accurately reporting our contributions. We provide a detailed discussion below regarding the reviewer’s concerns about the importance of the problem and its relevance in comparison to other approaches such as trajectory-based RL and RLHF in particular.
>
> > While I appreciate the contributions the authors made to CPT-based RL problems, I am not convinced of the importance of CPT-based RL problems by reading this paper. The paper cites another paper "... this is particularly important in applications directly involving humans in the loop such as e-commerce, crowdsourcing and recommendation to name a few" to argue that this class of problems has important applications.
>
> CPT is a popular model in behavioral economics originating from economics and psychology that was recognized by a Nobel prize attributed to Daniel Kahneman in 2002. We believe that developing a paradigm extending CPT to sequential decision making is an important and natural extension as it allows to broaden the scope of applications and factor in its main features by taking into account the subjective valuation of outcomes (utility) and the subjective weighting of probabilities. CPT-based RL unifies different settings including in particular risk-sensitive RL that has by now endless applications in RL.
> Please see our response below to your last comment for a detailed discussion regarding the importance of our problem and existing applications. We thank the reviewer for their comment, we will certainly add further motivation in the introduction to further support the importance of our problem. We have now added section C in the appendix p. 17 (due to space constraints) to discuss applications along the lines of the last comment below.
>
> > However, it is not clear from the cited paper whether other formulations that take into account human behavior can also handle these applications well. In fact, compared with CPT, I feel that a more principled way that considers human behavior is trajectory-based reward RL problems (one reward for the entire trajectory) with human preferences, like RLHF in large language models. While CPT assumes a structure of the final reward (the final metric to optimize involves a weight function, a utility function, and a summation of rewards), these weight functions are unknown and must be learned from data. In contrast, in trajectory-based reward RL problems, there is no assumption on the structure of the metric being optimized. And the metric is learned with human preference data. I wonder how authors think about the pros and cons of CPT in terms of its applications compared to trajectory-based reward RL problem settings.
>
> First of all, we do not exclude that other approaches might also be useful for modeling human behavior (and there are others as you mention), this is an open and promising research area that we believe has yet many interesting research directions to offer. We also do not claim that our approach is the unique best way to tackle the problem. Nevertheless, we do believe it is a principled approach rooted in an established literature in behavioral economics that would gain to be developed for sequential decision making. Our paper contributes to this effort by proposing a practical PG algorithm. While the CPT approach has not yet been well-established in the machine learning community, we believe it has interesting features to offer and it is already pervasive in RL through its particular cases risk-sensitive and safe RL. We provide a more detailed discussion below regarding these aspects and comment on your interesting question regarding the comparison with other paradigms such as RLHF/trajectory-based reward RL and the pros and cons of CPT.
>
> Please see the rest of our response to your above comment in what follows.

---

> > ### Author Response · Authors · 2024-11-21
> > **Response to reviewer aLSN (continued)**
> >
> > - Regarding the structure of the final reward and the metric learning you mention, this is a fair point and we agree that our work requires so far access to utility and weight functions. However, let us mention a few points:
> > 1. These can be readily available in specific applications (for risk modeling or even chosen at will by the users themselves);
> > 2. CPT relies on a predefined model, this can be beneficial in applications such as portfolio optimization or medical treatment where trade-offs have to be made and models might be readily available;
> > 3. Furthermore, we argue that having such a model allows it to be more explainable compared to a model entirely relying on human feedback and fine tuning, let alone the discussion about the cost of collecting human feedback. We also note that some of the most widely used algorithms in RLHF (e.g. DPO) do rely on the fact that the reward follows a Bradley-Terry model for instance (either for learning the reward or at least to derive the algorithm to bypass reward learning);
> > 4. Let us mention that one can also learn the utility and weight functions. We mentioned this promising possibility in our conclusion although we did not pursue this direction in this work. One can for instance represent the utility and weight functions by neural networks and train models to learn them using available data with relevant losses, jointly with the policy optimization task. One can also simply fit the predefined functions (say e.g. Tversky and Kahneman’s function) to the data by estimating the parameters of these functions (see $\eta$ in l. 162 with our notations and exponents of the utility function in table 1 p. 17 for the CPT row). This last approach is already commonly used in practice, see e.g. Rieger et al. 2017 (reference 1/ provided below in the discussion).
> >
> > - **CPT vs RLHF: General comparison.** CPT has been particularly useful when modeling specific biases in decision making under risk to account for biased probability perceptions. It allows to **explicitly** model cognitive biases. In contrast, RLHF has been successful in training LLMs which are aligned with human preferences where these are complex and potentially evolving and where biases cannot be explicitly and reasonably modeled. RLHF has been rather focused on learning **implicit** human preferences through interaction (e.g. using rankings and/or pairwise comparisons).  Overall, CPT can be useful for tasks where risk modeling is essential and critical whereas RLHF can be useful for general preference alignment although RLHF can also be adapted to model risk if human preferences are observable and abundantly available at a reasonable cost. This might not be the case in healthcare applications for instance, where one can be satisfied with a tunable risk model. On the other hand, so far CPT does not have this ability to adapt to evolving preferences over time unlike RLHF which can do so via feedback.
> >
> > - **CPT and RLHF: Pros and cons.** To summarize the pros and cons of both approaches, we provide the following elements. As for the pros, CPT directly models psychological human biases in decision making via a structured framework which is particularly effective for risk preferences. RLHF can generalize to different scenarios with sufficient feedback and handle complex preferences via learning from diverse human interactions, it is particularly useful in settings where preferences are not explicitly defined such as for LLMs for aligning the systems with human preferences and values. As for the cons, CPT is a static framework since the utility and probability weight functions are fixed, it is hence less adaptive to changing preferences. It uses a predefined model of human behavior which is not directly using feedback. It also requires to estimate model parameters precisely, often for specific domains. As for RLHF on the other hand, the quality and the quantity of the human feedback is essential and this dependence on the feedback clearly impacts performance. This dependence can also cause undesirable bias amplification which is present in the human feedback. We also note that training such models is computationally expensive in large scale applications.

---

> ### Author Response · Authors · 2024-11-21
> **Response continued**
>
> - **CPT and RLHF are not mutually exclusive.** While CPT and trajectory-based RL (say e.g. RLHF) both offer frameworks for incorporating human preferences into decision making, we would like to highlight that CPT and RLHF are not mutually exclusive. We can for instance use CPT to design an initial reward structure reflecting human biases, then refine it with RLHF. We can also consider to further relax the requirement of sum of rewards (which already has several applications on its own) and think about incorporating CPT features to RLHF. Some recent efforts in the literature in this direction that we mentioned in our paper include the work of Ethayarajh et al. (ICML 2024) ‘Model alignment as prospect theoretic optimization’ which combines prospect theory with RLHF (without probability weight distortion though, which limits its power). Note that the ideas of utility transformation and probability weighting are not crucially dependent on the sum of rewards structure and can also be applied to trajectory-based rewards or trajectory frequencies for instance. We believe this direction deserves further research, one interesting point would be how to incorporate risk awareness from human behavior to such RLHF models using ideas from CPT.
>
> Thank you for the interesting question. We have now incorporated this discussion to appendix D (p. 18, in the revised manuscript) due to space constraints.
>
>
> > Another weakness is related to the empirical study of the paper. The experiments only demonstrate the effectiveness of the proposed algorithm but do not include those justifying the necessity of the CPT problem. It would be much better if the authors could demonstrate that in some applications, the data shows that human behavior can be accurately predicted with CPT but not other theories.
>
> **About the importance of our problem formulation and its empirical relevance.** While our empirical results do not focus on the importance of our CPT problem formulation and mainly serve an illustrative purpose given our theoretical and algorithmic contributions to CPT-RL, we would like to stress that CPT has been tested and effectively used in a large number of compelling behavioral studies that we cannot hope to give justice to here. It is far from being an intellectual theoretical curiosity disconnected from practice. Besides the initial findings of Tversky and Kahneman for which the latter won the Nobel Prize in economics in 2002, please see a few recent references below for a broad spectrum of real-world applications  ranging from economics to transport, security and energy, mostly in the stateless (static) setting, also reflecting how active and impactful this research is in other fields. We quote some of the results for each paper for the convenience of the reader to reply to the reviewer’s concern. We hope these examples convince the reviewer of the importance of the problem we address which is an extension to the dynamical setting via our RL formulation for sequential decision making.
>
> 1. *Rieger, M. O., Wang, M., & Hens, T. (2017). Estimating cumulative prospect theory parameters from an international survey. Theory and Decision, 82, 567-596.*
> ‘We conduct a standardized survey on risk preferences in 53 countries worldwide and estimate cumulative prospect theory parameters from the data. The parameter estimates show that significant differences on the cross-country level are to some extent robust and related to economic and cultural differences. In particular, a closer look on probability weighting underlines gender differences, economic effects, and cultural impact on probability weighting.’
> Note here the explainability feature of CPT that we highlighted above in our discussion.
>
> 2. *Ebrahimigharehbaghi, S., Qian, Q. K., de Vries, G., & Visscher, H. J. (2022). Application of cumulative prospect theory in understanding energy retrofit decision: A study of homeowners in the Netherlands. Energy and Buildings, 261, 111958.* ‘CPT correctly predicts the decisions of 86% of homeowners to renovate their homes to be energy efficient or not. EUT, on the other hand, overestimates the number of decisions to renovate: it incorrectly predicts retrofit for 52% of homeowners who did not renovate for energy efficiency reasons. Using the estimated parameters of CPT, the cognitive biases of reference dependence, loss aversion, diminishing sensitivity, and probability weighting can be clearly seen for different target groups.’

---

> > ### Author Response · Authors · 2024-11-21
> > **Further examples**
> >
> > 3. *Gao, D., Xie, W., Cao, R., Weng, J., & Lee, E. W. M. (2023). The performance of cumulative prospect theory's functional forms in decision-making behavior during building evacuation. International Journal of Disaster Risk Reduction, 104132.*
> > ‘Understanding the performance of decision-making behavior in building evacuation is essential for predicting pedestrian dynamics, designing appropriate facility safety management, optimizing emergency management strategies, and reducing the impact of disasters. While many pedestrian movement models have been developed based on the hypothesis of rational and strategic decision-making, only a limited number of works consider individual psychology and irrational behavior. To address this issue, we have successfully integrated the cumulative prospect theory (CPT) into modeling evacuations.’
> >
> > 4. *Yan, Q., Feng, T., & Timmermans, H. (2020). Investigating private parking space owners’ propensity to engage in shared parking schemes under conditions of uncertainty using a hybrid random-parameter logit-cumulative prospect theoretic model. Transportation Research Part C: Emerging Technologies, 120, 102776.*
> > ‘Results show that socio-demographic characteristics, context variables, revenues and psychological concerns are all important factors in explaining parking space owners’ propensity to engage in platform-based shared parking schemes. [...] Understanding parking space owners’ propensity to share their parking spaces in relation to their psychological concerns and uncertain conditions is critical to improve shared parking policies. The results of this paper may help designers and planners in the delivery of shared parking services and promote the success and future growth of the shared parking industry.’
> >
> > 5. *Dorahaki, S., Rashidinejad, M., Ardestani, S. F. F., Abdollahi, A., & Salehizadeh, M. R. (2022). A home energy management model considering energy storage and smart flexible appliances: A modified time-driven prospect theory approach. Journal of Energy Storage, 48, 104049.* ‘Smart home is a small but an important energy segment that has a significant potential to implement authentic energy policies, where human is a major decision-maker in the home energy management dilemma. Therefore, humans’ emotions and tendencies plays a vital role as the End-User's daily decisions. In this paper, we develop a behavioral home energy management model based on time-driven prospect theory incorporating energy storage devices, distributed energy resources, and smart flexible home appliances. [...] The results of the simulation studies show that the End-User's satisfaction in the proposed home energy management behavioral structure will be increased substantially compared with the conventional monetary home energy management models.’
> >
> > 6. *Ladrón de Guevara Cortés, R., Tolosa, L. E., & Rojo, M. P. (2023). Prospect theory in the financial decision-making process: An empirical study of two Argentine universities. Journal of Economics, Finance and Administrative Science, 28(55), 116-133.* ‘This paper aims to provide empirical evidence for using the prospect theory (PT) basic assumptions in the Argentine context. Mainly, this study analysed the financial decision-making process in students of the economic-administrative academic area of two universities, one public and one private, in Córdoba. [...] The empirical results provided evidence on the effects of certainty, reflection and isolation in both universities, concluding that the study participants make financial decisions in situations of uncertainty based more on PT than on expected utility theory. This study contributes to the empirical evidence in a different Latin-American context, confirming that individuals make financial decisions based on the PT [...].’
> >
> > We thank the reviewer for their comment. We have now added a section in the appendix (see appendix C p. 17 in the revised manuscript, modifications all in blue) to further support the relevance of CPT compared to expected utility theory to model human behavior as suggested. We will also briefly update the introduction with the additional references and refer to the appendix for further details (due to space constraints). CPT provides a more realistic framework for modeling decision-making in uncertain environments than expected utility theory and this has been supported by extensive empirical studies. We believe that the machine learning community has a lot more to offer to address these questions given all the modern developments.
> >
> > Thanks again for your thoughtful review, please let us know if our response addresses your concerns. If you have any further questions or comments, we will be happy to address them.

---

> ### Comment · Reviewer_aLSN · 2024-12-03
> **My feedback and two follow-up questions**
>
> Your feedback on the usefulness of CPT-RL is appreciated. One suggestion is to make the example in your paper more concrete. Specifically, in the patient treatment example, you explained the potential benefits of weight functions, utility functions, and reference points. However, it would be more convincing if you could reference a study, if available, that models these functions and points using real patient treatment data.
>
> I have another question regarding the extension of algorithms beyond Policy Gradient (PG) to the CPT setting. Specifically, when it comes to learning a value function, do you think there are straightforward ways to adapt RL algorithms like TD learning or Q-learning to the CPT framework?
>
> Lastly, I'm a bit confused about your claim that "several risk measures are also particular cases of CPT values: Variance, Conditional Value at Risk (CVaR), distortion risk measures, to name a few." You also mentioned in the caption of Table 1 that "w+ and w− are often required to be continuous, which would exclude VaR and CVaR." These two statements seem conflicting—one suggests that CPT encompasses CVaR, while the other implies it does not. Could you clarify this?

---

> > ### Author Response · Authors · 2024-12-03
> > **Thank you for your feedback, our response to your follow-up questions**
> >
> > We sincerely thank the reviewer for their valuable feedback and suggestions which have helped improve our work.  We reply to their comments and follow-up questions in details in the following.
> >
> > 1. **About our healthcare example.** Thank you for the useful suggestion which we will follow, we agree that it will make the example even more compelling. There are indeed several behavioral studies in healthcare works using CPT and studying what you suggested. We will add them to support the example. We have found a number of recent such studies from which we cite a few together with some quotes to support our point:
> >
> > - **Mkrtchian, A., Valton, V., & Roiser, J. P. (2023). Reliability of decision-making and reinforcement learning computational parameters. Computational Psychiatry, 7(1), 30.** This is a psychological study  conducted on 50 participants recruited from the UCL Institute of Cognitive Neuroscience Subject Database and supported by the (British) National Institute for Health Research (NIHR). Here is a quote from the paper: ‘[...]risk aversion and loss aversion parameters from a prospect theory model exhibited good and excellent reliability, respectively. [...] These results suggest that reinforcement learning, and particularly prospect theory parameters, as derived from a restless four-armed bandit and a calibrated gambling task, can be measured reliably to assess learning and decision-making mechanisms. Overall, these findings indicate the translational potential of clinically-relevant computational parameters for precision psychiatry.’ The authors further add in the conclusion: ‘These models can further be used to predict future behaviour in the same individuals, especially PT model parameters, indicating that the decision-making processes assessed in these tasks represent relatively consistent and unique characteristics of an individual. These findings take us one step closer to translating computational measures of behaviour into clinical application.’
> >
> > - **George, S. A., Sheynin, J., Gonzalez, R., Liberzon, I., & Abelson, J. L. (2019). Diminished value discrimination in obsessive-compulsive disorder: A prospect theory model of decision-making under risk. Frontiers in Psychiatry, 10, 469.** This work studied decision-making in  a clinical cohort consisting of ‘patients diagnosed with OCD (n = 10), generalized anxiety disorder (n = 15), social anxiety disorder (n = 14), and healthy controls (n = 20) [which] were given a decision-making task and choices were modeled using a cumulative prospect theory framework.’ Here is what authors mention in their introduction: ‘Recently, behavioral neuroeconomic tools have been touted as having high potential utility in assessing the decision-making characteristics of people with psychiatric disorders (1, 8, 9). Such an approach computes “optimal” or normative behavior on a variety of dimensions, thus allowing for precise quantification of deviation from these norms. The traditional view conceptualizes decision-making as a rational process involving simple comparisons of expected values or expected utilities. However, because human behavior routinely deviates from purely “rational” choice, cumulative prospect theory offers empirically validated mathematical formulations of psychological effects in decision-making, such as loss aversion, and the circumstances when risk-seeking or risk-averse behaviors are likely to occur.’ Please refer to e.g. Figure 3 in their paper for estimated parameters of the CPT probability weighting and utility functions as well as further details regarding the results.
> >
> > - **Sip, K. E., Gonzalez, R., Taylor, S. F., & Stern, E. R. (2018). Increased loss aversion in unmedicated patients with obsessive–compulsive disorder. Frontiers in Psychiatry, 8, 309.** This is a study conducted on 43 obsessive–compulsive disorder patients across two sites (Icahn School of Medicine at Mount Sinai in New York and University of Michigan) supported by the US National Institutes of Health (NIH). ‘Obsessive–compulsive disorder (OCD) patients show abnormalities in decision-making and, clinically, appear to show heightened sensitivity to potential negative outcomes. Despite the importance of these cognitive processes in OCD, few studies have examined the disorder within an economic decision-making framework. Here, we investigated loss aversion, a key construct in the prospect theory that describes the tendency for individuals to be more sensitive to potential losses than gains when making decisions.’ The authors add: ‘These data identify abnormalities of decision-making in a subgroup of OCD patients not taking psychotropic medication. The findings help elucidate the cognitive mechanisms of the disorder and suggest that future treatments could aim to target abnormalities of loss/gain processing during decision-making in this population.’

---

> > > ### Author Response · Authors · 2024-12-03
> > > **response (continued)**
> > >
> > > Here a few additional relevant studies:
> > > - *Zhao, M., Wang, Y., Meng, X., & Liao, H. (2023). A three-way decision method based on cumulative prospect theory for the hierarchical diagnosis and treatment system of chronic diseases. Applied Soft Computing, 149, 110960.*
> > > - *Sun, J., Zhou, X., Zhang, J., Xiang, K., Zhang, X., & Li, L. (2022). A cumulative prospect theory-based method for group medical emergency decision-making with interval uncertainty. BMC Medical Informatics and Decision Making, 22(1), 124.*
> > >
> > > 2. **Question 1.** One has to be careful about such extensions because the nonlinearity of both the utility and probability weighting functions prevents from leveraging dynamic programming, especially under our (natural) formulation of the CPT-RL problem. There are a few works discussing value-based methods for CPT (e.g. Q-learning)  that we mentioned in our related works section (e.g. Borkar and Chandak 2021, Ramasubramanian et al. 2021). However, as we discuss it in related works, ‘these works are concerned with maximizing a sum of CPT value period costs which is amenable to dynamic programming. In contrast to their accumulated CPT-based cost (see their remark 1), our CPT policy optimization problem formulation is different: we maximize the CPT value of the return of a policy (see (CPT-PO)). In particular, this objective does not enjoy an additive structure and hence does not satisfy a Bellman equation.’
> > >
> > > **Question 2.** CVar, Var and distortion risk  measures are indeed CPT values **if we allow for discontinuous probability weighting functions**, we provided a short proof for our claim in appendix E.3 for completeness. It is  based on distortion risk measures from which VaR and CVaR have been shown to be special cases (Wirch & Hardy, 2001). To see this, for Var, we use a step function that focuses on the quantile. For CVaR, we use a piecewise uniform weighting function for the tail. The idea is that considering discontinuous weighting functions allows CPT to focus only on specific parts of the probability distribution which are relevant to Var and CVar. In general, probability weights and utility functions are supposed to be continuous as people might less frequently exhibit a behavior corresponding to hard thresholding probabilities. That was the point of the footnote. VaR and CVaR are rather rooted in the financial risk management literature and rely on objective probability distributions whereas CPT originates from behavioral economics and focuses on subjective risk perception and decision-making.
> > >
> > > The footnote in the caption is a bit misleading, we will update the formulation, the point of that footnote is to mention for completeness and to be completely rigorous that in those cases the probability weight functions are not continuous and this is a caveat. The point of our remark in the main part of the paper is to illustrate the modeling ability of the CPT framework in general.
> > >
> > > We suppose differentiability of the probability weight functions in our paper to be able to compute gradients everywhere for simplicity. Note though that non-differentiability of the weight functions is actually mild in those cases as it only occurs at a single point for Var and CVar, please see figure 6 in appendix E.2 (p. 19) where we represent these weight functions for Var and CVar for illustration purposes. To apply our first-order algorithm to non-smooth settings without worrying about this single point issue for example, one can consider smooth approximations of these distortion functions (we did discuss and test such an alternative, see appendix G.2 for details). One can also further consider Clark subdifferentials to address the nonsmooth setting, we stick to the differentiable case which already allows us to consider a wide range of utility and weighting functions for simplicity and clarity of the paper. We note though that there are better approaches to address the specific case of CVar in terms of optimization (by fully leveraging its structure, see e.g. Meng, S. Y., & Gower, R. M. A model-based method for minimizing CVaR and beyond. ICML 2023, for an approach based on subgradients as we highlight and a more sophisticated one using a stochastic prox linear method, see also Duchi and Ruan 2018 or Davis and Drusvyatskiy 2019). We highlight that our main goal is to make use of the features of general CPT and not just to generalize prior risk measures which do not capture all the important features of CPT that we have expanded on. As a side note, we also do not claim that our method should be uniformly the best algorithm to use for all settings (as one might often exploit specific structures of special problems). We hope this clarifies your confusion, we will further clarify this in the paper.
> > >
> > > We hope this answers your questions precisely. Please let us know if you have any further questions, we will be happy to address them.

---

> > > > ### Comment · Reviewer_aLSN · 2024-12-03
> > > >
> > > > I see. In this case, I feel that it would be better by not restricting to continuous weight functions when you define CPT values. Then you can say that CPT values encompass CVar, Var, etc. Then when you introduce the method, you could say that the method only works for continuous weight functions. But discontinuous functions are probably not that important to consider. Plus, our method has advantages A, B, and C.
> > > >
> > > > I don't have more questions.

---

> > > > > ### Author Response · Authors · 2024-12-03
> > > > > **Thank you**
> > > > >
> > > > > Thank you for your response. Let us mention that we have clearly indicated the continuity and differentiability assumptions in the statement of our policy gradient theorem (Theorem 6). We will surely make this clearer as for the special cases like CVar and Var, thank you for the suggestions regarding this that we will follow.

---

### Official Review · Reviewer_hWfL · 2024-11-01

**Soundness:** 2
**Presentation:** 2
**Contribution:** 2
**Rating:** 5
**Confidence:** 3

**Summary:**

This paper is a follow-up work to [L.A. et al., 2016] on Cumulative Prospect Theory (CPT) value optimization in reinforcement learning problems. The original paper utilized the simultaneous perturbation stochastic approximation (SPSA) method to update the policy, while this paper provides a policy gradient method. The policy gradient is evaluated and compared with the SPSA method in several domains.

**Strengths:**

+ Discussion on the optimal policies under CPT, showing that optimal policies are generally stochastic and non-Markovian when CPT is applied.
+ Policy gradient algorithm for CPT value optimization, compared to SPSA methods in the original paper.

**Weaknesses:**

+ The algorithm is only validated in a few small domains, making it difficult to assess the performance of this policy gradient in more complex tasks.
+ The computation of the integral for the CPT value appears complex and time-consuming due to the use of Monte Carlo sampling.

Suggestions:

+ The authors may consider demonstrating the benefits of using CPT over standard risk-neutral RL in a concrete reinforcement learning problem at the beginning of the paper to help readers better understand CPT.

+ This paper considers the problem in a non-discounted setting, i.e., $\gamma=1$ as shown in Line 181. The claim in line 261 does not apply to the discounted case for exponential utilities. Discussion may be needed. see [Entropic Risk Optimization in Discounted MDPs, AISTATS 2023]

+ As mentioned in line 309, Proposition 7 in the paper is different from the Proposition 6 in [L.A. et.al. 2016], authors may need to provide a comparison and justification.

**Questions:**

+ $\phi(R(\tau))$ in line 295, 303, 305 should be $\varphi(R(\tau))$ in Theorem 6?

+ How to choose the proper utility function for different problems?

+ The policy gradient calculation depends on the property that the function $u$ is non-decreasing. Can we work with other $u$ where non-decreasing is not guaranteed?

---

> ### Author Response · Authors · 2024-11-21
> **Rebuttal: response to reviewer hWfL**
>
> We thank the reviewer for their time and feedback. We answer their questions and reply to their comments in the following.
>
> > The algorithm is only validated in a few small domains, making it difficult to assess the performance of this policy gradient in more complex tasks.
>
> Besides our small scale grid world experiments, please note that we did also test the performance of our PG algorithm on a continuous state action space setting with our electricity management scenario. In practice, prior work in CPT-RL (L.A. et al. 2016, ICML) did only consider a SPSA (zeroth order) algorithm (which we also compared to) in a traffic signal control application on a 2x2-grid network. Our PG algorithm has clearly an advantage compared to a zeroth order algorithm which only relies on function values and which is harder to scale to larger state action spaces. **We have now performed additional simulations during the rebuttal phase for a financial trading application** (see response to reviewer Ydtj, results and precise description in the last two pages of our revised manuscript, all modifications in blue).
>
> > The computation of the integral for the CPT value appears complex and time-consuming due to the use of Monte Carlo sampling.
>
> Monte Carlo sampling is used even for the vanilla policy gradient algorithm (say Reinforce) for standard expected return RL. We use a similar procedure here which is simple. Our additional required quantile estimation procedure requires a mild sorting step which can be executed in $O(n \log n)$ running time (without even invoking parallel implementations) where $n$ is the length of the rewards to be sorted.
>
> > Suggestion: The authors may consider demonstrating the benefits of using CPT over standard risk-neutral RL in a concrete reinforcement learning problem at the beginning of the paper to help readers better understand CPT.
>
> We have provided concrete simple examples of CPT in the introduction to give the reader some intuition without introducing any notation nor background. For clarity, we prefer to defer a detailed exposition of examples of CPT-RL to later in the paper (simulation section) once the problem formulation is precisely exposed and explained. Thank you for the suggestion, we will add a pointer to the introduction to directly refer the reader to the relevant section for an example.
>
> > This paper considers the problem in a non-discounted setting, i.e., γ=1 as shown in Line 181. The claim in line 261 does not apply to the discounted case for exponential utilities. Discussion may be needed. see [Entropic Risk Optimization in Discounted MDPs, AISTATS 2023].
>
> - We thank the reviewer for their comment and for the reference. Indeed, we focus on the undiscounted setting in our work. This is precisely because some of the issues discussed in the paper you mention arise when considering discounted rewards (e.g. lack of positive homogeneity). The entropic risk measure (ERM) mentioned in the reference you provide seems to be actually rather consistent with our result (Theorem 4), up to the monotone log function applied to the objective (which does not fundamentally change the policy optimization problem). Comparing (5) and (8) in the reference with our (EUT-PO) problem formulation, ERM is rather consistent with the exponential form we provide in Theorem 4 (up to the fact that our theorem does not apply to the discounted setting as we currently state it). As for the entropic value-at-risk (which builds on the entropic risk measure), it seems that it is not exactly an instance of our problem formulation (EUT-PO) because of the supremum over $\beta$ transformation coupling ERM with the additive log term (besides the discounted setting).
>
>
> - Although discounting is widely adopted in the RL community, especially for infinite horizon settings, we do not find the finite horizon to be a restrictive setting in practice in applications. As for the extension to the infinite horizon, we find the use of positive quasi-homogeneity to adapt the proof and show the existence of an optimal deterministic Markov policy (which is crucially non-stationary) for the special case of Entropic Risk Measure (Theorem 3.3 therein) interesting. This observation might actually give some hope to extend our result to the discounted setting using similar techniques.
> We will add a discussion regarding this interesting point, thanks again for your comment and the useful reference that we were not aware of.

---

> > ### Author Response · Authors · 2024-11-21
> > **Response to remaining comments and questions**
> >
> > > As mentioned in line 309, Proposition 7 in the paper is different from the Proposition 6 in [L.A. et.al. 2016], authors may need to provide a comparison and justification.
> >
> > We provided a brief comment about this difference in l. 343-346. As we mention it, the reason for this is that the policy gradient theorem (Theorem 6) involves the exact same integral term (l. 288) that we approximate in Proposition 7 (l. 319) and this term features a first order derivative of the weight function since we are considering gradients. L. A. et al. 2016 do not require such derivative in the integral because they only consider zeroth order estimates of the policy gradient. Overall, the approximation result is fundamentally the same as the Riemann scheme approximation of the integral using simple staircase functions does not crucially depend on the integrand ($w’$ in our case, $w$ in their case) which we replace for our purpose.
> >
> > > $\phi(R(\tau))$ in line 295, 303, 305 should be $\varphi(R(\tau))$ in Theorem 6?
> >
> > Thank you for spotting the typo. It is corrected now.
> >
> > > How to choose the proper utility function for different problems?
> >
> > - The problems themselves might dictate to the user or decision maker the utility function to be used. The user might also design their own according to their own beliefs, behaviors and objectives, based on the goal to be achieved (e.g. risk-seeking, risk-neutral, risk-averse). Specific applications might also suggest specific utility functions such as specific risk measures like in risk sensitive RL for instance. We have provided in table 1 p. 17 a list of different examples one might consider. Learning the utility function is also an interesting direction to investigate as we mention in the conclusion. In practice, it is rather common to use the example we provide in table 1 p. 17 (CPT row) with exponent parameters which are estimated using data.
> >
> > - We provide a few concrete examples in the following. For instance, Rieger et al. 2017 (see reference below in response to reviewer aLSN) adopt such an approach (see sections 3.1, 3.2 and 3.3 therein for a detailed discussion about parameter estimation). Ebrahimigharehbaghi et al. 2022 (see reference below) choose some similar variation of this utility (see eqs. 2-3 therein) while still using KT’s probability weighting functions. Gao et al. 2023 compare different functions for different similar power utility functions with fitted parameters (see Tables 1, 2 and 3 therein p. 3, 4, 6 for extensive comparisons with the existing literature). Similar investigations were conducted in Yan et al. 2020. Dorahaki et al. 2022 consider psychological time discounted utility functions (variations of the same power functions)  in their model with additional relevant hyperparameters, motivated by (domain-specific) psychological studies (see eq. (4) therein). It is worth noting that all these examples are only in the static stateless setting.
> >
> > We thank the reviewer for the question, we will add a remark along these lines to our paper about this point.

---

> ### Author Response · Authors · 2024-11-21
> **Response to 3rd question**
>
> > The policy gradient calculation depends on the property that the function u is non-decreasing. Can we work with other u where non-decreasing is not guaranteed?
>
> - Strict monotonicity is needed in some of the proofs of our results, mainly Theorem 4, see e.g. proof of 1 implies 2 in p. 24 and proof of 5 implies 2 in p. 22. However, it is not formally required to derive our policy gradient theorem. Note also that even Proposition 7 for estimating the integrals does not require it as soon as $\xi_{i/n}^{+,-}$ denote the right quantiles as we defined them in the proposition. See point 3 below for further comments.
>
> - We focus on the monotone setting because typical human behavior tends to prefer better outcomes over worst ones, e.g. higher gains over smaller ones and smaller losses over larger losses. This is captured by the monotonicity assumption on the utility function: utility increases with increasing gains and decreases with increasing losses. This is a fundamental assumption in economics and decision theory which is consistent with how humans evaluate outcomes. Nevertheless, there are cases where this assumption might not hold if this is what the reviewer is referring to.
>
> - Mere monotonicity is mainly useful in our algorithm to guarantee that the utility values are sorted in the same order as the returns obtained from the sampled trajectories (see step 6 of the algorithm for quantile estimation computation). This leads to a simple computation of the quantiles of the utilities: Once the returns are sorted, the utilities are also sorted in the same way and quantiles can be read from the sorted list. If monotonicity does not hold anymore, one has to be more careful about this computation and adapt step 6 of the algorithm accordingly by simply computing the quantiles of the utilities using the right sorting of these (which would be different from the sorting of the returns). In that case, sorting the returns in step 5 is not needed and one only needs to sort the utilities (utility function applied to the returns). Apart from this technical detail, we do not foresee any major impediment to using the algorithm without the strict monotonicity assumption.
>
> - We thank the reviewer for this question, we will add a remark to the paper accordingly.
>
> Thank you for your review, please let us know if you have any further concern or questions.

---

> > ### Comment · Reviewer_hWfL · 2024-11-23
> >
> > Thanks for the detailed response! I have the same feeling as reviewer aLSN that I did not capture the importance of CPT-based RL after reading the paper. The advantage of CPT is that it is a generalization, while the drawback is that it is not intuitive to interpret. The paper provides a very brief example in the introduction, which does not adequately motivate the use of CPT. The authors may consider addressing questions such as how to choose the utility function and what the CPT represents when choosing a specific utility, to help readers better understand the CPT-RL problem.
> >
> > In addition, the experimental domains are relatively simple. Many risk RL papers conduct experiments in Mujoco (e.g., [1, 2, 3, 4]). The authors may consider designing more complex experiments to validate the proposed algorithm.
> >
> > [1] Risk-Averse Offline Reinforcement Learning, ICLR 2021
> > [2] Mean-variance policy iteration for risk-averse reinforcement learning, AAAI 2021
> > [3] An alternative to variance: Gini deviation for risk-averse policy gradient, NeurIPS 2023
> > [4] One risk to rule them all: A risk-sensitive perspective on model-based offline reinforcement learning, NeurIPS 2023

---

> ### Author Response · Authors · 2024-11-28
> **Additional example and experiment**
>
> We thank the reviewer for their feedback. We address their concerns in the following. Regarding the importance of CPT-RL, we refer the reviewer to our detailed response to reviewer aLSN and appendix C p. 17 which we added to the manuscript. We have also augmented the introduction with l. 63-68 to highlight applications. Concerning examples motivating CPT-RL, we provide another example below and we have now added it to the main part of the paper in section 2 p. 4-5 to illustrate our CPT-RL problem formulation as suggested by the reviewer (deferring related work to the appendix due to space constraints). As for the utility function choice, we will add a discussion following our response to your question above.
>
> **Additional concrete example of CPT-RL in healthcare: Personalized Treatment for Pain Management.** CPT is not just a generalization, its features are important in applications, especially when human perception and behavior matter. Existing traditional RL approaches often lack a behavioral perspective and CPT-RL allows for more empathetic and realistic  sequential decision making. Here is another concrete example to illustrate the importance of CPT-RL and its differences compared to risk sensitive RL to provide more intuition to the reader.
>
> -**Scenario**: The goal is to help a physician manage a patient's chronic pain by suggesting a personalized treatment plan over time. The challenge here is to balance pain relief and the risk of opioid dependency or other side effects that might be due to the treatment, i.e. short-term relief and longer term risks.
>
> -**Our approach**:  We propose to train a CPT-RL agent to help the physician.
>
> **Why sequential decision making?**
>
> (a) The physician needs to adjust treatment at each time step depending on the patient’s reported pain level as well as the observed side effects. Note here that this is relevant to dynamic treatment regimes in general (such as for chronic diseases) in which considering delayed effects of treatments is also important (and RL does account for such effects). We refer the reader to section IV of Yu and Liu 2020, ‘Reinforcement Learning in Healthcare: A Survey.’ ACM Computing Surveys.
>
> (b) Decisions clearly impact the patient's immediate pain relief, dependency risks in the future and their overall health condition.
>
> **Why CPT?** Patients and clinicians make decisions influenced by psychological biases. We illustrate the importance of each one of the three features of CPT as introduced in our paper in section 2 (reference point, utility and probability distortion weight functions) via this example:
>
>  (a) *Reference points*: Patients assess and report pain levels according to their subjective (psychologically biased) baseline. Incorporating reference point dependence leads to a more realistic model of human decision-making as this allows for capturing e.g. expectations, past experience as well as their desired outcomes to define their perceived gains and losses. In our example, reducing pain from a level of 7 to 5 is not perceived the same way if the reference point of the patient is 3 of it is 5. In contrast, risk-sensitive RL treats every pain reduction as a uniform gain, regardless of the patient’s starting reference pain level.
>
> (b) *Utility transformation*: Patients might often show a loss averse behavior, i.e., they might perceive pain increase or withdrawal symptoms as worse than equivalent gains in pain relief. **Note here that loss aversion should not be confused with risk aversion** (see definition and discussion in Schmidt and Zank 2005. ‘What is loss aversion?’ The Journal of Risk and Uncertainty.) In short, loss aversion can be defined as a cognitive bias in which the emotional impact of a loss is more intense than the satisfaction derived from an equivalent gain. For instance, in our example, a 2-point increase in pain might be seen as much worse than a 2-point reduction even if the change is the same in absolute value.  This loss aversion concept is a cornerstone of Kahneman and Tversky’s theory. In contrast, risk aversion rather refers to the **rational** behavior of undervaluing an uncertain outcome compared to its expected value. Risk sensitive approaches might be less adaptive to a patient’s subjective preferences if they deviate from objective risk assessments.
>
> (c) *Probability weighting*: Low probability events such as severe side effects (e.g., opioid overdose or dependency) might be overweighted or underweighted based on the patient's psychology.
>
> **Environment and transitions.** A state is a vector of three coordinates (current pain level, dependency risk, side effect severity). Actions are treatments, e.g. no treatment, alternative treatment or opioid treatment. An episode ends if the patient develops full dependency or if pain is effectively managed.

---

> ### Author Response · Authors · 2024-11-28
>
> **CPT-RL vs Risk-averse RL: comparison.**
> In terms of policies, risk-averse RL would favor non-opioid treatments unless extreme pain levels make opioids justifiable. In contrast, CPT-RL policies would prescribe opioids if pain significantly exceeds the patient’s reference point. As dependency risk increases, CPT-RL policies would transition to non-opioid treatments as a consequence of overweighting the probability of rare catastrophic outcomes. Notably, CPT-RL policies can oscillate between risk-seeking (to address high pain) and risk-averse (to avoid severe side effects). In contrast,  a risk-sensitive agent focuses on minimizing variability in health states and dependency risks and would likely avoid opioids in most cases unless pain levels become extreme. Such risk-sensitive policies favor stable strategies (e.g., consistent non-opioid use), prioritizing low variance in patient outcomes.
>
> **Experiments.** In our simulations, we have focused on examples emphasizing the behavioral economics motivation of CPT-RL. As for Mujoco, we have **now also added an example on the inverted pendulum environment to demonstrate that our PG algorithm can be readily used in such environments as well (see page 38 in the appendix).**

---

### Official Review · Reviewer_Ydtj · 2024-11-03

**Soundness:** 3
**Presentation:** 2
**Contribution:** 2
**Rating:** 3
**Confidence:** 3

**Summary:**

The paper introduces a novel approach to reinforcement learning (RL) by leveraging Cumulative Prospect Theory (CPT) to account for human decision-making biases, moving beyond the traditional expected utility framework. It focuses on the policy optimization problem, where the objective is the CPT value of the random variable recording the cumulative discounted rewards (CPT-PO). Additionally, it considers the particular case of the expected utility objective, where only returns are distorted by the utility function (EUT-PO). This work derives a policy gradient (PG) theorem for CPT-based RL, presenting a model-free policy gradient algorithm for optimizing a CPT policy.

**Strengths:**

This paper addresses an objective that encompasses a broad class, including Conditional Value at Risk (CVaR), distortion risk measures, and expected utility.

The paper effectively explains various policy classes associated with different objectives, helping to clarify how these classes relate to CPT-RL versus standard RL approaches.

The derivation of a policy gradient theorem for CPT objectives is a significant extension of the traditional PG results in reinforcement learning, broadening its applicability in human-centered decision contexts.

**Weaknesses:**

The experimental section provides limited insight due to its use of relatively straightforward examples (traffic control and electricity management), which may not fully illustrate the complexities that arise in more realistic, high-stakes environments, such as finance or healthcare.

In Section 5(a), the observation that different weight functions lead to different policies in the grid environment could be strengthened by assessing these policies against risk measures (such as CVaR) beyond expected return. Without this, it is unclear if the CPT-RL-PO policy outperforms standard RL-PO under any specific risk-sensitive criteria.

The grid environment results are overly simplified and provide little substantive information. It would be beneficial to evaluate whether the derived policies' performance aligns meaningfully with their respective objectives.

Limited applicability: The experiments did not demonstrate the advantages of using the proposed algorithm. It is unclear what the benefits of using CPT learning are over risk-sensitive RL or distributional RL. Although CPT is theoretically valuable, the empirical advantages of CPT over other risk-sensitive measures remain ambiguous in the presented results. The authors may consider including direct comparisons with risk-sensitive RL or distributional RL methods on the same tasks.

**Questions:**

Minor comment: On page 4, under Problem formulation: CPT-RL, the cumulative discounted rewards variable $X$ is referenced, but the definition provided does not discount the rewards. It would be clearer to either update the variable's definition to include discounting or adjust the notation accordingly to prevent confusion.

---

> ### Author Response · Authors · 2024-11-21
> **Rebuttal: response to Reviewer Ydtj**
>
> We thank the reviewer for their feedback. Please find below a point by point reply to your comments and concerns.
>
> >  'The experimental section provides limited insight due to its use of relatively straightforward examples (traffic control and electricity management), which may not fully illustrate the complexities that arise in more realistic, high-stakes environments, such as finance or healthcare.'
>
> **Traffic control and electricity management** are major high-stakes and complex applications which are extremely active research areas. Such applications need no motivation regarding their complexity and relevance in the real world. As a matter of fact a quick keyword search on app.dimensions.ai shows more than 200 000 (resp. more than 100 000) publications around traffic control (resp. electricity management) in 2023.
>
> **Our simulations.** While our simulations rely on simplified models that cannot capture all the real-world intricacies of such complex applications, our goal is to illustrate how our problem formulation can be useful to solve concrete problems by giving a flavor about the scenarios one might consider in practice. We purposefully designed them to go beyond the standard vanilla gridworld that we have also considered. In particular, please note that we are using real-world public data available online for our simple electricity management application (please see appendix F. 6, Figure 19). We report electricity prices on a typical day on the market together with the total electricity production.
>
> Besides our theoretical contributions which we would like to highlight, please let us briefly recall the purpose of each one of the applications we consider and the insights we provide: (a) in our traffic control application, we show the influence of the probability distortion function, (b) we illustrate the scalability of our PG algorithm to larger state spaces compared to the existing zeroth order algorithm in a grid environment with increasing state space size and (c) we test our algorithm in a continuous state-action space setting in our electricity management application.
>
> **Additional simulations.** Finance and healthcare are also important applications. We thank the reviewer for mentioning these. Exploring all these applications in depth is beyond the scope of this paper. Nevertheless, **during the rebuttal period we have now performed additional simulations in a financial application as requested by the reviewer**. The goal is to train RL trading agents using our general PG algorithm in the setting of our CPT-RL framework. We use a Gym Trading environment available online using data from the Bitcoin USD (BTC-USD) market between May 15th 2018 and March 1st 2022. One can easily use any other real-world publicly available stock market data by providing a dataframe as input to build a corresponding RL environment. Our BTC-USD data is of the same kind as any historical data which can be found online (see e.g. https://finance.yahoo.com/quote/BTC-USD/history). The RL trading agent can take three classical positions (SHORT, OUT and LONG). We refer the reader to the last pages of our revised manuscript (modifications all in blue) for the figure as well as more details regarding the experiment. We hope these additional simulations further convince the reviewer about the practicality of our algorithm.
>
> > In Section 5(a), the observation that different weight functions lead to different policies in the grid environment could be strengthened by assessing these policies against risk measures (such as CVaR) beyond expected return. Without this, it is unclear if the CPT-RL-PO policy outperforms standard RL-PO under any specific risk-sensitive criteria.
>
>  Please note that we do consider a risk averse probability weight function to compare to the risk neutral case. We refer the reader to Table 2 p. 28, appendix F. 5 p. 31 for an illustration of the probability weight function w^+ used and Figure 18 for representations of the policies obtained that illustrate that the output policies solving our PO problem are meaningful compared to standard RL-PO.
>
> > The grid environment results are overly simplified and provide little substantive information. It would be beneficial to evaluate whether the derived policies' performance aligns meaningfully with their respective objectives.
>
> - Our experiment with a varying grid world state space size illustrates the better scalability of our PG algorithm compared to the previously known zeroth order algorithm (see Figure 3).
> - We did investigate the policies that we obtain by solving our problem using our novel PG algorithm, see appendix F.4, figure 16 p. 31 for representations of the policies we obtain and how they are consistent with the objectives set. See also figure 18 p. 32.
> - We have now performed additional simulations to apply our methodology and use our PG algorithm in financial trading (see last part of the revised manuscript for a detailed exposition).

---

> ### Author Response · Authors · 2024-11-21
> **About applicability**
>
> > Limited applicability: The experiments did not demonstrate the advantages of using the proposed algorithm. It is unclear what the benefits of using CPT learning are over risk-sensitive RL or distributional RL. Although CPT is theoretically valuable, the empirical advantages of CPT over other risk-sensitive measures remain ambiguous in the presented results. The authors may consider including direct comparisons with risk-sensitive RL or distributional RL methods on the same tasks.
>
> **Our general PG theorem result and our novel proposed algorithm**  expand the applicability of CPT-RL in practice besides being theoretically grounded. We would like to highlight that our PG theorem unifies several settings including standard RL, risk sensitive, risk seeking, probability distorted settings and beyond under the umbrella of CPT-RL. Our algorithm which stems from such a general result is therefore general as for the different problem settings it can address. We refer the reader to p. 17 for a diagram representing our framework in the literature and Table 1 for different special case examples.
>
> **Advantage and applicability.** We devoted a specific part of our experiments to clearly illustrate the scalability advantage of our PG algorithm compared to the zeroth order algorithm (please see sec. 5. (b) and fig. 3, the performance of the 0th order algorithm gets even worse with an even larger scale). We also demonstrated its applicability to different settings including the case of continuous state action settings, different utility functions including risk seeking and risk averse ones, and different probability weight functions as well.
>
> **Comparison to Risk sensitive RL.** Risk sensitive RL is a particular case of our framework and our goal is not to outperform existing algorithms for specific tasks using our general algorithm. We also provided several examples where we use risk-sensitive measures though. We also note that prior work in risk-sensitive RL (e.g. Vijayan and L. A. AISTATS 2023 that we mention in related works) has also used similar SPSA algorithms as the one we compare to in section 5. (a). Moreover, particularizing our problem and algorithm to the special case of the risk-sensitive setting with exponential criteria results in an algorithm that bears similarity to existing work and PG algorithms such as the work of Noorani et al. 2023 and we refer the reader to this paper for comparisons to other risk-sensitive algorithms for that particular task. Distributional RL focuses on modeling and computing the entire distribution of returns. While distributional RL allows handling risk explicitly by approximating the distribution of returns, it is in general computationally demanding and it does not model subjective risk handling via utility and probability distortions like CPT.
>
> **Comparison to CPT.**
> - CVar or VaR in risk sensitive RL do not take into account transformed utilities (modeling perceived/subjective gain and loss values differently and wrt reference points) whereas exponential risk sensitive RL does not account for probability weighting (which models low and high probability events subjective perception). To appreciate this, please see Table 1 in p.17 and our experiments where we compare the different examples (please see figure 2).
>
> - When an agent perceives a return differently from the raw return, then this is also modeled in our CPT framework and it is clear that CPT is more suitable than existing risk sensitive approaches. This is the case when (a) it has a different reference point regarding their perceived value, (b) its utility is not linear in the return, it is rather concave for gains and convex for losses (wrt their reference).
> - Most importantly, we stress that our CPT framework captures several existing risk sensitive measures as particular cases and offers the flexibility to the decision maker to design their own. In particular, CPT can at the same time overweight low probability events and underweight highly probable ones with different magnitudes. Therefore, CPT allows to handle losses and gains separately. A similar flexibility is offered for utility functions. See figure 6 p. 18 for illustrations.

---

> > ### Author Response · Authors · 2024-11-21
> > **About our theoretical contributions and response to minor point**
> >
> > **About our theoretical contributions.** Besides our algorithmic contributions and experiments, we would like to draw the attention of the reviewer to our theoretical contributions. We are not aware of any work which establishes a PG theorem in this level of generality. This result on its own is conducive to a number of possible algorithmic schemes from which we propose a vanilla PG algorithm (one can think about actor-critic methods, n-step methods, variance reduced schemes and many others). We investigate the  nature of optimal policies on our problem and show the differences with respect to the standard RL setting. In particular, we characterize the utility functions allowing for Markovian policies (which are sufficient policies in standard RL but not necessarily in our setting) and we show that they reduce to the class of affine and exponential utilities. We are not aware of any such result in the literature.
> >
> > > **Minor comment:** On page 4, under Problem formulation: CPT-RL, the cumulative discounted rewards variable X is referenced, but the definition provided does not discount the rewards. It would be clearer to either update the variable's definition to include discounting or adjust the notation accordingly to prevent confusion.
> >
> > Thank you for spotting this. Throughout the paper, we focus on the finite horizon undiscounted setting corresponding to the formulation of page 4 and the definition of X in the same page. We comment on the extensions to the discounted and infinite horizon settings in remarks 8 (l. 887) and 13 (l. 1373). This is fixed now on page 4.
> >
> > We thank the reviewer again for their time and feedback. Please let us know if you have any further comment or questions, we will be happy to address them.

---

> > > ### Comment · Reviewer_Ydtj · 2024-12-02
> > >
> > > Thank you to the authors for the responses and the additional experiments.
> > >
> > > Some concerns remain. The authors stated that “several risk measures are also particular cases of CPT values: Variance, Conditional Value at Risk (CVaR), distortion risk measures, to name a few.” It would be worthwhile to demonstrate that the proposed algorithm is at least comparable to existing risk-sensitive RL algorithms in some particular cases, such as under the popular risk metric CVaR. This comparison will help ensure that the learned CPT policy’s performance aligns meaningfully with their respective objectives. For example, in the financial trading and MuJoCo experiments, it is unclear how the proposed algorithm performs compared to existing risk-sensitive RL algorithms.
> > >
> > > Also, the paper is difficult to follow. The writing and presentation of the manuscript need improvement to make the content clear and accessible to readers. I will maintain the current score.

---

> > > > ### Author Response · Authors · 2024-12-03
> > > > **Response to the reviewer's comments**
> > > >
> > > > Thank you for your feedback.
> > > >
> > > > - **Our motivation and goal.** Please note that our goal is not to design an algorithm to compare to existing special risk measures which are studied in the literature. CPT-RL is our problem formulation and it has many interesting features that are not captured by other existing distortion risk measures (as we have discussed in our previous response) which also have their own merits depending on the application and the goal the agent pursues. Our main motivation is to address the CPT-RL problem and showcase its importance. We do not believe that comparing to existing risk-sensitive RL algorithms for specific risk-sensitive objectives brings an added value in that it ‘will help ensure that the learned CPT policy’s performance aligns meaningfully with their respective objectives’ as the reviewer suggests. Comparing to other algorithms for special cases of our problem (which do not emphasize or even consider the motivation of CPT-RL and its features) does not enhance our main motivation in this work.
> > > > - **About our experiments.** We have tested our algorithm in a number of applications, going far beyond prior work in CPT-RL which only considered solving a single traffic control application on a 2x2 grid using a zeroth order algorithm. We have investigated their alignment with the objective we pursue, investigating even the policies we obtain. Our additional experiments explore the sensitivity to different reference points as well as different parameters of the CPT model to show how CPT return is affected, including in risk-sensitive variants of the problem. **In the financial training and MuJoCo experiments we considered, existing risk-sensitive RL algorithms cannot be applied to our setting in which we consider both probability weight and utility functions (which are not both the identity, nor they are any step function such as the ones needed to model Var or CVar) such as the KT model.**
> > > > - **Instantiation of our PG algorithm for exponential criteria.** Particularizing our problem and algorithm to the special case of the risk-sensitive setting with exponential criteria essentially results in an algorithm which is the PG algorithm proposed in Noorani et al. 2023 and we refer the reader to this paper for comparisons to other risk-sensitive algorithms for that particular risk-sensitive task. Our work focuses on showcasing the benefits of CPT-RL objectives which were not previously studied in practice.
> > > > - **Comparison to SPSA which was also investigated in risk-sensitive RL.** We also note that prior work in risk-sensitive RL (e.g. Vijayan and L. A. AISTATS 2023 that we mention in related works) has also used similar SPSA algorithms as the one we compare to in section 5. (a) (for CPT-RL) and we did compare to such an algorithm to show the advantage of our method.
> > > >
> > > > > ‘Also, the paper is difficult to follow. The writing and presentation of the manuscript need improvement to make the content clear and accessible to readers. I will maintain the current score.’
> > > >
> > > > **Presentation.** Regarding the presentation, we would like to highlight that reviewer aLSN (in its original review) finds that **‘the paper’s presentation is clear’**, ‘the theoretical and algorithmic contributions are original and **clear**. If one believes that CPT-based RL problems are important, then these contributions are also significant' and that ‘the paper did a very nice job relating its studied problem with other related problem settings studied previously, **making it easy for readers to understand the position of this paper in the literature**.’ Please let us know if you have any concrete and constructive comment to further improve our presentation and we will be happy to take it into account. During the rebuttal, we have also made substantial efforts to revise the manuscript and strengthen it, please see our general comment for a summary of our revisions where we also highlight our contributions and their importance.

---

### Author Response · Authors · 2024-11-29
**Revision and contributions**

We would like to thank the reviewers again for their feedback and their valuable suggestions.

**Revision.** In addition to our detailed responses to each one of the reviewers, we have uploaded a revised version of our manuscript containing the following main modifications (highlighted in blue) to improve our work based on the reviewers’ feedback which we are grateful for:
- Regarding the importance of CPT-RL (highlighted by reviewers aLSN and hWfL), we have updated the introduction with l. 63-68 to highlight applications of CPT in general and we have incorporated a new extended discussion in appendix C p. 17.
- Concerning examples to clarify the features of CPT-RL and its comparison to risk-sensitive RL, we have added another concrete example in healthcare (personalized treatment for pain management) to section 2 right after the problem formulation to illustrate it and highlight why we consider CPT-RL instead of prior existing risk-sensitive approaches. We will further update the manuscript to incorporate more details in the lines of our response to reviewer hWfL.
- We have added appendix D p. 18-19 to compare CPT-RL and trajectory-based reward RL as preference based learning paradigms to address the questions and concerns of reviewer aLSN.
- We deferred the related work section to appendix B due to space constraints.
- **Experiments:** In addition to our simulations in grid worlds, simple traffic control settings as well as our electricity management application using real-world data, we have added simulations for trading in financial markets (appendix H. 7, as suggested by reviewer Ydtj) and on the Mujoco inverted pendulum environment (appendix H. 8, as suggested by reviewer hWfL) to show the applicability of our PG algorithm to other settings.

**Our key contributions.** Our main goal in this work is to study and solve a policy optimization problem accounting for human perception and behavior via the use of the well-established cumulative prospect theory from behavioral economics. We provide further insights on the optimal policies and their nature in such problems compared to standard RL policy optimization for maximizing expected returns. We would like to highlight that one of our main contributions is our PG theorem and algorithm to solve this problem. We believe that our practical PG algorithm is widely applicable. We have tested it in several applications, **going far beyond prior work in CPT-RL which only considered solving a single traffic control application on a 2x2 grid using a zeroth order algorithm**. Testing it in other applications and even larger scale problems would certainly be interesting. Our work also opens up the way to several interesting future work opportunities towards more realistic human-based sequential decision making both from the algorithmic and methodological viewpoints (such as learning reference points, utilities and probability weight functions as we highlight in our paper and in our responses to reviewers).

**Potential.** We are excited by the potential that CPT-RL has to offer to better model human sequential decision making and incorporate cognitive and psychological biases which are of most importance in several high-stakes applications. Only few works have been devoted to this endeavour in RL and our work contributes to this goal.  We have only scratched the surface in this regard by showcasing the potential of solving such a CPT-RL problem formulation in simple settings for healthcare decision making, energy, finance and investment. Many other meaningful applications such as legal and ethical decision making, cybersecurity and human-robot interaction are yet to be explored.

We would be happy to address any further questions or concerns during the remainder of the discussion period.

---

### Meta-Review · Area_Chair_UHSq · 2024-12-19

**Metareview:**

This paper is on policy optimization when the objective is Cumulative Prospect Theory (CPT) value of the cumulative reward random variable. It proves a policy gradient theorem for CPT-RL and derives a policy gradient algorithm for optimization in CPT-RL. Finally, it empirically shows the efficacy of the proposed algorithm and compares it with an existing method for optimizing CPT-RL that is based on simultaneous perturbation stochastic approximation (SPSA).

(+) The policy gradient theorem, the corresponding algorithm, and experimenting with several problems are all positive aspects of this work.

(-) Given that CPT-RL was previously studied by L.A. Prashanth et al. (2016), the reviewers expected that this paper provides a better comparison, in terms of both theoretical and empirical results, with the existing work.
(-) The reviewers seem to have concerns about the motivation behind the CPT formulation in RL. I believe the authors need to better justify the importance of this formulation in RL.
(-) There are several questions about the connections between the CPT-RL formulation and the proposed policy gradient algorithm, and the variety of risk-sensitive RL formulations and algorithms (many of them being policy gradient) that exist in the literature. I believe addressing this issue can significantly improve the overall quality of the paper.

**Additional Comments On Reviewer Discussion:**

The authors tried to address the issues raised by the reviewers and answer their questions. Although some of the questions were addressed, the reviewers still had some concerns, especially on topics that were summarized in the meta-review. I believe the paper has potential and I strongly recommend that the authors take the reviewers' comments into account, improve the quality of their work, and submit it to an upcoming venue.

---

### Decision · Program_Chairs · 2025-01-22

Reject